# Probing Rotary Position Embeddings through Frequency Entropy

**Yui Oka**[1][*], **Kentaro Hanafusa**[2][*][†], **Taku Hasegawa**[1], **Kyosuke Nishida**[1], **Kuniko Saito**[1]
[1]Human Informatics Labs., NTT, Inc.
[2]Ehime University
yui.oka@ntt.com

## Abstract

Rotary Position Embeddings (RoPE) are widely used in Transformers to encode positional information in token representations, yet the internal frequency structure of RoPE remains poorly understood. Previous studies have reported conflicting findings on the roles of high- and low-frequency dimensions, offering empirical observations but no unifying explanation. In this paper, we present a systematic framework that bridges these disparate results. We introduce Frequency Entropy (FE), a metric that quantifies the effective utilization of each RoPE frequency dimension, and we provide an analysis of how RoPE's sinusoidal components contribute to model representations on a per-dimension basis. Based on an analysis of the Llama-4 model, which incorporates both RoPE and NoPE layers, we find that the periodicity captured by FE appears in RoPE layers but not in NoPE layers. Furthermore, FE identifies dimensions in which energy concentrates under RoPE. These characteristics are observed across the spectrum rather than being confined to specific dimensions. Moreover, attenuating extreme-entropy dimensions at inference yields downstream accuracy that is statistically indistinguishable from the baseline, with modest perplexity improvements on average, suggesting that such dimensions are often redundant. Overall, FE provides a simple, general diagnostic for RoPE with implications for analysis and design.

## 1 Introduction

Position representations in Transformers (Vaswani et al., 2017) are a crucial factor determining their ability to handle long-range dependencies. Among these representations, Rotary Position Embeddings (RoPE) (Su et al., 2023) have become standard in many of the latest large language models, including Llama (Touvron et al., 2023; Grattafiori et al., 2024), Qwen (Qwen et al., 2025; Yang et al., 2025), and Gemma (Gemma Team et al., 2024a), and have contributed to improved performances in long-text processing. However, the design of RoPE was introduced empirically, and the role of each frequency dimension and how they are utilized within the model remain unclear.

In recent years, several analyses at the RoPE dimension level have been reported. For example, Barbero et al. (2025) observed that high-frequency components contribute to positional pattern formation, while low-frequency components contribute to semantic information. They also demonstrated that replacing part of the low-frequency components with NoPE (Kazemnejad et al., 2023) does not significantly affect model performance. On the other hand, Chiang & Yogatama (2025) showed that high-frequency components are scarcely utilized and can be removed without impacting performance. Furthermore, Hong et al. (2024) pointed out that the low-frequency component is essential for modeling long-range dependencies in specific attention heads. Previous analyses of RoPE have largely relied on visual inspection of heatmaps and coarse division into high- versus low-frequency components. The Llama-4 (Meta, 2025) model introduces *iRoPE*, which combines interleaved NoPE layers with frequency scaling to extend context length, though its internal frequency dynamics remain largely unexplored. These examples highlight the conflicting reports regarding the

---

[*]These authors contributed equally.
[†]Work done during internship at NTT, Inc.

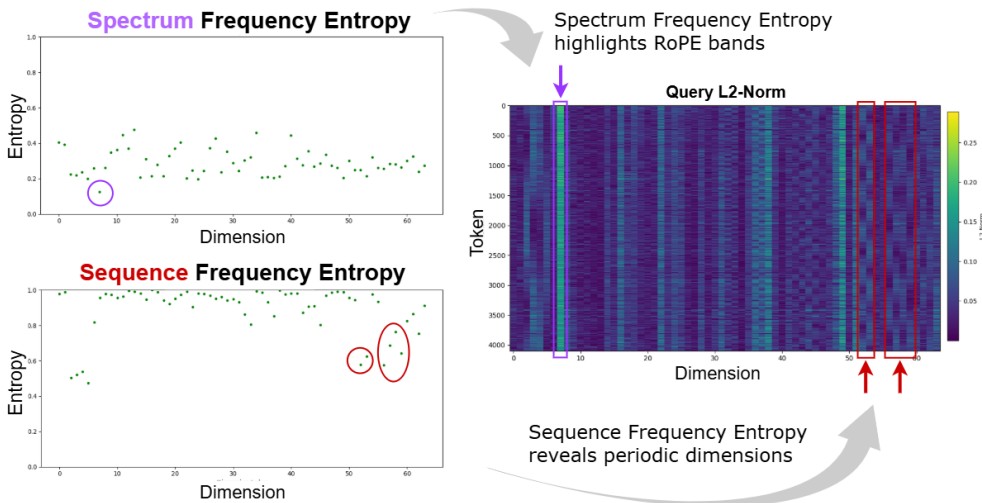

Figure 1: Example of our Frequency Entropy. Left: Frequency Entropy (FE) per rotary pair, where the horizontal axis is the entropy value and the vertical axis is the pair index $j \in \{0, \ldots, d/2 - 1\}$. Right: Query $\ell_2$-norms across rotary pairs, where the horizontal axis is sequence length ($L = 4096$) and the vertical axis is the pair index.

role of each RoPE dimension, and analysis remains observation-based, lacking a unified theoretical and empirical understanding.

In this paper, we mathematically formalize the contribution of each frequency dimension and introduce **Frequency Entropy (FE)**, a classical quantitative measurement framework. It comprises two complementary metrics, Spectrum Frequency Entropy (Spectrum FE) and Sequence Frequency Entropy (Sequence FE), and makes RoPE's spectral behavior measurable on a per-dimension basis. Figure 1 illustrates the method. We compute these entropies on a Llama-4 model with RoPE and NoPE layers and find that Spectrum FE quantifies power concentration in the Fourier domain and reveals band-limited rotation pairs with sustained energy, while Sequence FE measures token-wise regularity and detects persistent oscillations of rotation pairs induced by RoPE. Furthermore, these signatures are absent in layers with NoPE.

To probe the functional relevance without fine-tuning, we conduct a targeted attenuation study. For rotation pairs whose FE falls below a threshold, we reduce their contribution on attention during inference by multiplying the corresponding query and key channels by a constant $\alpha < 1$, keeping all other parameters fixed. The experimental results demonstrate that suppressing low-Sequence-FE dimensions leaves perplexity and downstream accuracy unchanged, whereas suppressing low-Spectrum-FE dimensions worsens perplexity. In addition, attenuating high-Spectrum-FE outliers has a negligible effect on the perplexity and the downstream accuracy. These results indicate that persistent oscillations induced by RoPE are largely redundant, while band-limited components carry a task-relevant signal.

Across experiments, we observe that oscillatory and band-limited behaviors occur throughout the spectrum rather than only at specific frequencies or extremes. Consequently, the conventional high-versus low-frequency dichotomy is insufficient. FE provides a spectrum-aware, model-agnostic lens that reconciles prior mixed observations and informs pruning, reweighting, and the design of future positional schemes.

## 2 BACKGROUND AND RELATED WORK

### 2.1 ROTARY POSITION EMBEDDING (ROPE)

RoPE (Su et al., 2023) has become the de facto standard positional embedding method in many of the latest large language models, such as Llama (Touvron et al., 2023; Grattafiori et al., 2024) Qwen (Qwen et al., 2025; Yang et al., 2025), and Gemma (Gemma Team et al., 2024a). RoPE

introduces position information by applying a rotation to the query and key vectors in the self-attention mechanism. This property allows RoPE to encode relative positional information while preserving compatibility with an absolute token index.

$$A_{m,n} = (R_{n,\theta}q_n)^{\top}(R_{m,\theta}k_m) = q_n^{\top} R_{m-n,\theta} k_m \tag{1}$$

The rotation matrix $R_{n,\theta} \in \mathbb{R}^{d \times d}$ is the block-diagonal rotation matrix, defined as follows:

$$R_{n,\theta} = \begin{bmatrix} \cos(n\theta_0) & -\sin(n\theta_0) & 0 & 0 & \dots & 0 & 0 \\ \sin(n\theta_0) & \cos(n\theta_0) & 0 & 0 & \dots & 0 & 0 \\ 0 & 0 & \cos(n\theta_1) & -\sin(n\theta_1) & \dots & 0 & 0 \\ 0 & 0 & \sin(n\theta_1) & \cos(n\theta_1) & \dots & 0 & 0 \\ \vdots & \vdots & \vdots & \vdots & \ddots & \vdots & \vdots \\ 0 & 0 & 0 & 0 & \dots & \cos(n\theta_{d/2-1}) & -\sin(n\theta_{d/2-1}) \\ 0 & 0 & 0 & 0 & \dots & \sin(n\theta_{d/2-1}) & \cos(n\theta_{d/2-1}) \end{bmatrix} \tag{2}$$

where $A \in \mathbb{R}^{L \times L}$, $q_n \in \mathbb{R}^{1 \times d}$ is the $n$-th query when the number of dimensions is $d$ and $n$ is the absolute position ($0 \leq n \leq L-1$) when the sequence length is $L$, and $k_m \in \mathbb{R}^{1 \times d}$ is the $m$-th key ($0 \leq m \leq L-1$). Typically, the rotation angles are chosen as the base of RoPE $\theta_j = 10000^{-2j/d}$ ($j = 0, \dots, \frac{d}{2}-1$). In practice, the base values $\theta$ in RoPE are typically set to be quite large. For example, $\theta = 10{,}000$ is adopted in the Gemma model (Gemma Team et al., 2024a) and Llama-2 (Touvron et al., 2023), $\theta = 500{,}000$ is used in Llama-3 (Xiong et al., 2024), and $\theta = 1{,}000{,}000$ is employed in Qwen-3 (Yang et al., 2025). RoPE provides a complex phase concept of relative offsets for attention across all layers and all heads while remaining parameter-free and hardware-friendly.

## 2.2 UTILIZATION OF INFORMATION WITHIN RoPE

The distribution and utilization of information within RoPE at the dimensional level remain only partially understood. An early perspective at the attention-head level emphasized the low-frequency structure as necessary for long-distance modeling. Hong et al. (2024) identified "positional heads" whose activations align with token distance. Ablating these heads—dominated by lower-frequency (slower-varying) RoPE dimensions—substantially degrades the long-context performance. A second line of empirical evidence dissected pretrained models to associate roles with high and low frequency. (Barbero et al., 2025) reported that high-frequency components drive distinctive off-diagonal "positional" attention patterns, whereas low-frequency components correlate more with semantic content. Further, replacing parts of the low-frequency spectrum with NoPE-like (Kazemnejad et al., 2023) variants leaves performance largely intact or even improved in small-to-mid-scale settings. These results suggest redundancy within portions of the low-frequency subspace. A third line of work has argued almost the converse for the high-frequency end: Chiang & Yogatama (2025) measured the per-dimension utilization and showed that dimensions with larger rotation angles (i.e., higher frequencies) are weakly used. This points to an over-allocation of representational capacity to rapid positional oscillations that downstream layers seldom exploit.

Methodologically, existing analyses have tended to employ broad classifications such as high-frequency/low-frequency, lacking a unified metric that transcends dimensions. Taken together, these findings create a tension: (i) particular heads critically depend on low-frequency structure for long-range dependencies (Hong et al., 2024); (ii) parts of the low-frequency spectrum appear semantically entangled and sometimes dispensable (Barbero et al., 2025); yet (iii) high-frequency dimensions look under-utilized and safely removable (Chiang & Yogatama, 2025).

## 2.3 IRoPE

Llama-4 (Meta, 2025) introduces *iRoPE*, which augments RoPE with two key changes. First, it utilizes an interleaved layer design where rotary position embeddings are applied only to alternating attention layers, while the others operate without explicit positional signals (NoPE; Kazemnejad et al., 2023), encouraging content-based long-range reasoning. Second, iRoPE scales the RoPE rotation angles by a factor $\alpha < 1$, slowing phase growth and extending the effective positional range well beyond the training context. Despite these advances, the internal behavior of iRoPE remains largely unexplored.

## 2.4 FREQUENCY ENTROPY

Frequency entropy (also called *spectral entropy*) quantifies the uncertainty of a signal's frequency distribution by applying Shannon's entropy (Shannon, 1948) to its power spectrum. Let $P(f_i)$ be the power spectral density at the $i$-th discrete frequency bin, where $i = 1, \ldots, N$ and $N$ is the total number of frequency bins obtained from the discrete Fourier transform. We normalize the spectrum to obtain a probability mass function $p_i = \frac{P(f_i)}{\sum_j P(f_j)}$. The frequency entropy is calculated as

$$H = -\sum_i p_i \log_2 p_i. \tag{3}$$

When desired, $H$ can be normalized by $\log_2 N$ to yield $0 \leq H \leq 1$. A low value of $H$ indicates that the spectral energy is concentrated in a few frequencies and is therefore highly ordered—for example, as in a pure tone—whereas a high value indicates that the energy is spread across many frequencies and is therefore more disordered, as in white noise. Thus, frequency entropy provides a single quantitative measure of the flatness or peakiness of a spectrum. This metric is widely used in signal processing and information theory (Misra et al., 2004; Sucic et al., 2014).

## 3 FREQUENCY ENTROPY: A NEW LENS FOR ROTARY POSITION EMBEDDINGS

After RoPE, each pair is rotated by an angle $n\theta_{d/2-1}$ with $\theta_{d/2-1}$ determined by the RoPE base. We are interested in *how strongly* each rotary pair exhibits narrow-band, RoPE-driven periodicity along the sequence, as opposed to content-driven, broadband variability.

**Core idea.** For each rotary pair, we construct a 1D block observable along the token axis and measure its frequency entropy (FE), defined as the Shannon entropy of the power spectrum. To quantify the utilization of each dimension, we introduce two new evaluation metrics: **Spectrum Frequency Entropy (SpectrumFE)** and **Sequence Frequency Entropy (SequenceFE)**. SpectrumFE evaluates which frequency components are present at any given moment. SequenceFE evaluates how periodic or irregular the energy fluctuations are in each dimension. Our FE is computed as follows:

1. Split the query into $d/2$ RoPE blocks and compute the $\ell_2$-norm of each, treating the resulting length-$L$ vector as a discrete signal (Section 3.1).

2. Compute two variants of power spectrum and applying Shannon entropy (Shannon, 1948) to its power spectrum as shown in Eq. (3) (Sections 3.2 and 3.3).

Low entropy indicates that the time series is dominated by a small number of frequencies, while high entropy indicates spectrally diverse, content-modulated dynamics. This yields a per-pair score that is model-agnostic, scale-free, and comparable across layers, heads, and architectures.

## 3.1 FROM ROPE BLOCKS TO A DISCRETE FREQUENCY SIGNAL

To measure the usage of frequencies, we start by noting any Cauchy-Schwarz equalities following (Barbero et al., 2025). The effect of the $j$-th frequency component on the activation $A_{n,m}$ is upper bounded by the $\ell_2$-norm of the query and key components, i.e., $\left|\langle q_n^{(j)}, k_m^{(j)}\rangle\right| \leq \|q_n^{(j)}\|_2 \|k_m^{(j)}\|_2$ $(j = 0, \ldots, \frac{d}{2} - 1)$. For a fixed block $j$, we collect these vectors across the sequence by

$$\mathbf{s}_j := \left[ \|q_0^{(j)}\|_2, \|q_1^{(j)}\|_2, \ldots, \|q_{L-1}^{(j)}\|_2 \right]^\top \in \mathbb{R}^L. \tag{4}$$

where $L$ is the sequence length. Therefore, we assume that the set of $\ell_2$-norms $\|q_n^{(j)}\|_2$ of queries after RoPE constitutes a discrete signal. We calculate the frequency entropy of the $d/2$ discrete signal patterns and measure the frequency utilization rate. In practical terms, measuring the $\ell_2$-norm of a query after RoPE is natural: the rotation aligns the representation with the frequency blocks actually used in the logit and preserves norms, so $\|q_n^{(j)}\|_2$ is both position-invariant and directly interpretable. We treat Eq. (4) as a discrete-time signal.

## 3.2 Spectrum Frequency Entropy

We measure the spectral complexity of this sequence using normalized spectral entropy. For a given signal $\mathbf{s}_j$, first compute the short-time Fourier transform (STFT) to obtain the power spectrum, as

$$S_{k,t} = \left| \sum_{n=0}^{F-1} \mathbf{s}_j[tH + n]\, w[n]\, e^{-i\frac{2\pi}{F}kn} \right|^2. \tag{5}$$

where $t(t = 0, 1, 2, \ldots, T-1)$ is each frame, $w[n]$ is the analysis window, and $S_{k,t}$ is the power spectrum at frequency bin $k(k = 0, 1, 2, \ldots, K-1)$. We set the frame length $F$ to 1024, hop length $H$ to 512, and sequence length $L$ to 4096. Therefore, the frequency bin is $K = \frac{F}{2} + 1 = 513$ and number of frames $T = \lfloor \frac{L-F}{H} \rfloor + 1 = 7$. Second, the power spectrum for each frame is normalized to form a probability distribution, as

$$p_k = \frac{S_k}{\sum_{j=0}^{K-1} S_j}, \quad S_k = \frac{\sum_{t=0}^{T-1} S_{k,t}}{T}. \tag{6}$$

Next, the Shannon entropy $H$ (Shannon, 1948) is calculated as

$$H = -\sum_{k=0}^{K-1} p_k \, \log_2 p_k. \tag{7}$$

To obtain a scale-free measure, we divide by the maximal entropy $H_{\max} = \log_2 K$, yielding the normalized spectral entropy $\tilde{H} = \frac{H}{H_{\max}}$. The normalized spectral entropy $\tilde{H}$ represents the Spectrum Frequency Entropy (SpectrumFE), quantifying the temporal spectral diversity of the query $\mathrm{L}\ell_2$-norm signal. Finally, since RoPE organizes the embedding into $\frac{d}{2}$ two-dimensional rotary pairs, we compute $\tilde{H}_j$ for each pair index $j \in 0, \ldots, \frac{d}{2} - 1$. Hence, the entropy is defined along $j$ and yields a length-$\frac{d}{2}$ vector $(\tilde{H}_0, \ldots, \tilde{H}_{\frac{d}{2}-1})$, with one value per rotary pair.

## 3.3 Sequence Frequency Entropy

For a given signal $\mathbf{s}_j$, first compute the discrete Fourier transform (DFT) to obtain the power spectrum as follows:

$$S_k = |\sum_{n=0}^{L-1} \mathbf{s}_j[n]\, e^{-i\frac{2\pi}{L}kn}|^2, \qquad k = 0, 1, \ldots, L-1. \tag{8}$$

Next, discard the DC component ($k = 0$) and restrict to the positive frequencies $1 \le k \le \frac{L}{2} - 1$. Define the total positive–frequency energy and normalize to obtain a probability distribution over these frequencies:

$$p_k = \frac{S_k}{\sum_{k=1}^{\lfloor L/2 \rfloor - 1} S_k}, \qquad k = 1, \ldots, \lfloor \frac{L}{2} \rfloor - 1. \tag{9}$$

Finally, the Shannon entropy $H$ (Shannon, 1948) is calculated as

$$H = -\sum_{k=1}^{\lfloor L/2 \rfloor - 1} p_k \, \log_2 p_k. \tag{10}$$

To obtain a scale-free measure, we divide by the maximal entropy $H_{\max} = \log_2 K$, yielding the normalized spectral entropy $\tilde{H} = \frac{H}{H_{\max}}$. The normalized spectral entropy $\tilde{H}$ represents the Sequence Frequency Entropy (SequenceFE), quantifying the temporal spectral diversity of the query $\mathrm{L}\ell_2$-norm signal. Finally, since RoPE organizes the embedding into $\frac{d}{2}$ two-dimensional rotary pairs, we compute $\tilde{H}_j$ for each pair index $j \in 0, \ldots, \frac{d}{2} - 1$. Hence, the entropy is defined along $j$ and yields a length-$\frac{d}{2}$ vector $(\tilde{H}_0, \ldots, \tilde{H}_{\frac{d}{2}-1})$, with one value per rotary pair.

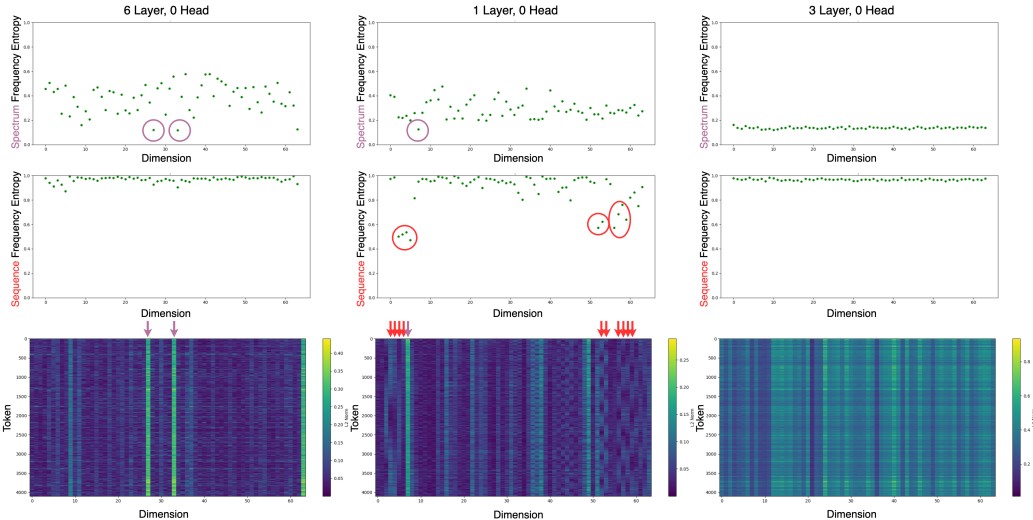

Figure 2: Scatter plots of each FE value in the `Llama-4-Scout-17B-16E-Instruct` model. Columns: layer 6, layer 1, layer 3 (left to right). Rows: SpectrumFE, SequenceFE, and query $\ell_2$-norm map. Top/middle: pair index in RoPE $j$ (x) vs. normalized entropy $\tilde{H}_j$ (y). Bottom: $j$ (y) vs. token index $n$ (x); color denotes $\|q_n^{(j)}\|_2$. All results are shown for head 0. The pair index $j$ is rotation patterns. Sequence length $L = 4096$.

## 4 ANALYSIS VIA FREQUENCY ENTROPY

In this section, we investigate the characteristic behavior of RoPE. To isolate their contribution, we compare queries that employ RoPE with those using no positional encoding (NoPE), enabling a direct identification of RoPE-specific effects. For this purpose, we primarily analyze the Llama-4 model with *iRoPE*, which alternates RoPE and NoPE across attention layers. [1]

### 4.1 SETTINGS

We conduct experiments using the `Llama-4-Scout-17B-16E-Instruct` (Meta, 2025) model with a head dimension of 128 and RoPE with 64 rotation patterns, 48 transformer layers, and 40 attention heads. In the `Llama-4-Scout-17B-16E-Instruct` model, a total of $(64 \times 48 \times 40) = 122,880$ frequency-entropy values are computed. For evaluation, we sample text at random from the `wikitext-103-raw-v1` split of the WikiText-103 dataset (Merity et al., 2017) and then concatenate the samples to form sequences of exactly 4096 tokens. Each such sequence is passed to the model, and we extract the attention query vectors during inference. We then compute each frequency entropy from these queries.[2]

### 4.2 ANALYSIS RESULTS

**What characteristics does SpectrumFE capture?** In Fig. 2, the top panel shows scatter plots of SpectrumFE per rotary pair, and the bottom panel shows the corresponding query $\ell_2$-norm maps. In the query $\ell_2$-norm map of layer 6, we observed contiguous stretches of rotary-pair indices with persistently elevated norms. We refer to contiguous ranges of rotary-pair indices that exhibit persistently high query $\ell_2$ norms as *frequency bands*. Similar banded patterns were also reported by Barbero et al. (2025) and are especially evident in positional heads. Dimensions with the smallest SpectrumFE align with pronounced band-limited patterns in the $\ell_2$-norm maps, indicating that

---

[1]The analysis of keys is in Appendix A, and the layer-wise analysis is in Appendix B. We also conduct the same analysis on models that apply RoPE in all layers and heads, including Llama-3, Qwen3, and Gemma2. See Appendix C for details.

[2]We additionally provide a context-length analysis in Appendix F a cross-dataset analysis in Appendix F.2, and an ablation study comparing the RoPE-only and NoPE-only models in Appendix G.

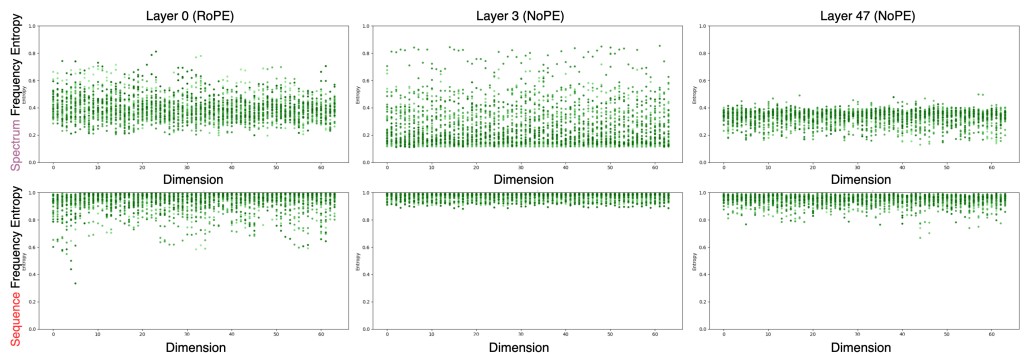

Figure 3: Scatter plots of FE values across all heads in the `Llama-4-Scout-17B-16E-Instruct` model. Columns: layer 0, layer 3, layer 47 (left to right). Layer 0 is the RoPE layer, while the others are NoPE layers. Rows: entropy scatter plots for SpectrumFE and SequenceFE. Scatter plots show pair index in RoPE $j$ (x) vs. normalized entropy $\tilde{H}_j$ (y). We vary the color intensity of the scatter points by head. The pair index $j$ is rotation patterns.

SpectrumFE captures the frequency-band structure. Across RoPE-applied layers, SpectrumFE values predominantly lie in the range 0.2–0.6. The SpectrumFE distribution in the NoPE layer (layer 3) yields a markedly different profile from the RoPE layers. In the NoPE layer (layer 3), multiple frequency bands are observed in the query $\ell_2$-norm maps.

**What characteristics does SequenceFE capture?**    In Fig. 2, the middle panel presents scatter plots of SequenceFE per rotary pair. In layer 0 (RoPE layer), pairs with the smallest SequenceFE exhibit clear periodic oscillations in the corresponding query $\ell_2$-norm maps, whereas in layer 3 (NoPE layer), no periodic pattern is observed and SequenceFE values rarely fall below 0.8. These observations indicate that SequenceFE is sensitive to periodic structure along the token axis. Across RoPE-applied layers, SequenceFE predominantly ranges from 0.2 to 0.6. In contrast, in the NoPE layer (layer 3), no periodicity emerges and the SequenceFE distribution concentrates near 1.0, yielding a markedly different profile from the RoPE layers (layers 0 and 6).

**Effect of the NoPE Layer**    The bottom row of Fig. 2 shows query $\ell_2$-norm heatmaps indicating that RoPE layers and the NoPE layer at layer 3 exhibit markedly different behavior. From SpectrumFE, the shallow NoPE layer (layer 3) exhibits more frequency bands than the RoPE layer. Meanwhile, from SequenceFE, the NoPE layer does not show rotating pairs and therefore does not exhibits clear periodic oscillations.

**Entropy Landscapes Across Layers and Heads**    Fig. 3 shows scatter plots of each FE per rotary pair in all heads. In the SpectrumFE plot, the shallow NoPE layer (layer 3) exhibits more dimensions with frequency-band-like characteristics than the RoPE layer. Furthermore, the SpectrumFE distribution is widely scattered. However, as the layer deepens (final layer 47), the number of dimensions with band-like characteristics decreases, and the SpectrumFE distribution converges within a certain range. In the SequenceFE plot, the shallow RoPE layer (layer 0) indicates that there are several periodic dimensions. However, in the NoPE layer, this periodic dimension disappears for all heads, regardless of whether the layer is shallow or deep. These results suggest that NoPE may attenuate the periodic structure characteristic of RoPE and emphasize frequency bands.

## 4.3    Why SpectrumFE shows bands and SequenceFE shows periodicity?

In summary, our analysis separates two types of structure in RoPE-driven attention: Spectrum Frequency Entropy reveals band-focused allocation across rotary pairs, and Sequence Frequency Entropy reveals tokenwise periodicity. NoPE may weaken the latter while preserving or highlighting the former. Deep layer may reduce both band sharpness and periodicity.

**Why SpectrumFE shows bands?**    SpectrumFE takes a short-time spectrum of the query-norm for each rotary pair by STFT and measures the Shannon entropy over frequency bins. SpectrumFE

measures how narrowly concentrated the local frequency content (via STFT) is. Entropy is low when energy sits in a few bins and high when it is spread out. Therefore, plotting low SpectrumFE across indices (bins) forms a contiguous low-entropy region, i.e., a frequency band. Low SpectrumFE indicates strong frequency-band structure, meaning the model consistently allocates energy to specific rotary pairs.

**Why SequenceFE shows periodicity?**   SequenceFE uses a global Fourier spectrum along the token axis by DFT and measures the entropy over positive frequencies. SequenceFE measures global periodicity of the RoPE-transformed signal (via DFT). Low SequenceFE indicates near–single-tone oscillation driven by RoPE's fixed rotational phase, rather than content. Low entropy means a simple, near-single-tone signal, and high entropy means a complex or noise-like signal. With RoPE active, a rotary pair advances at an almost constant step per token, so the query oscillates at a fixed frequency. If we remove RoPE, the fixed-rate oscillation vanishes, energy spreads, and SequenceFE rises.

**Why bands can persist in NoPE for SpectrumFE?**   Even without rotational oscillation, the model may up-weight some rotary pairs due to content or architectural bias. This uneven allocation still concentrates energy for those indices within short windows, keeping SpectrumFE low over a contiguous set of indices. At the same time, the per-token signal is not strongly periodic, so SequenceFE remains high.

## 5   FILTERING OUT OUTLIER DIMENSIONS

Low SpectrumFE reflects band-limited behavior, while low SequenceFE reflects strong periodicity. The following question arises: *Are frequency bands and periodicity redundant elements for the model, or are they essential components?* In this section, guided by Frequency Entropy, we intervene in RoPE by selectively weighting these rotary pairs to mitigate their contribution.

### 5.1   WEIGHTED RoPE

If Frequency Entropy is below a certain threshold, we weight the corresponding RoPE dimension to reduce its effect. We call this *Weighted RoPE*. Let $\tilde{H}^{(l,h,j)} \in [0,1]$ denote the Frequency Entropy in the query for layer $l$, head $h$, and rotary pair $j$. Given a threshold $\tau \in (0,1)$ and a RoPE weight $\alpha \in [0, 0.1, ..., 0.9]$, we set a weighted factor as follows:

$$\alpha^{l,h,j} = \begin{cases} \alpha, & \text{if } \tilde{H}^{l,h,j} < \tau, \\ 1, & \text{otherwise.} \end{cases} \tag{11}$$

We modulate the RoPE transformation applied to the $j$-th rotary pair of the query at token position $m$. Let $R_{m,\theta}^{(l,h,j)} \in \mathbb{R}^{2 \times 2}$ denote the standard RoPE rotation for pair $j$ in layer $l$ and head $h$. We obtain the weighted query subvector by scaling the usual rotation with $\alpha^{l,h,j}$, which acts as a soft mask that attenuates the contribution of low-entropy pairs while leaving others unchanged, as

$$q_m^{(j)\star} = \alpha^{l,h,j} R_{m,\theta}^{(l,h,j)} q_m^{(j)} \tag{12}$$

Intuitively, for low-entropy pairs, we slow the phase growth and attenuate periodicity; for other pairs, RoPE remains unchanged. We calculate the FE for each key and perform the same processing.

### 5.2   PERPLEXITY

**Settings**   Four publicly available large language models were used for the evaluation: Llama-4-Scout-17B-16E (Meta, 2025), gemma-2-9b-it (Gemma Team et al., 2024b), Qwen3-8B (Yang et al., 2025), and Meta-Llama-3-8B (Grattafiori et al., 2024). We evaluated the inference perplexity on the test set of the wikitext-103 dataset (Merity et al., 2017). For the threshold parameter $\tau$, we adopted different settings according to the entropy metric. For Spectrum Frequency Entropy, we examined two regimes: one where $\tau$ was greater than 0.4 and one where $\tau$ was less than 0.2. For Sequence Frequency Entropy, we observed that outliers occurred only at low values, so we evaluated the case where $\tau$ was less than 0.6. No fine-tuning was performed in any of the experiments.

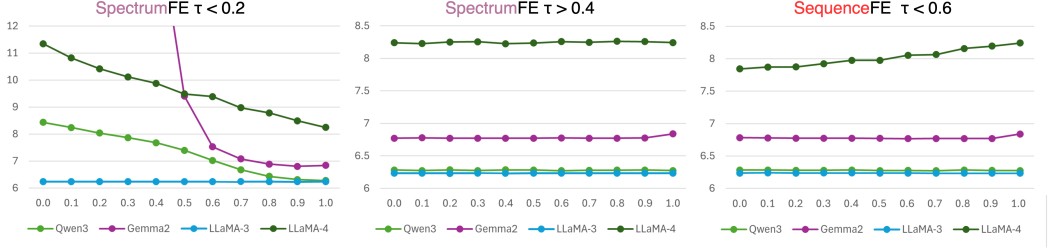

Figure 4: Perplexity as a function of the RoPE weight $\alpha$ under entropy-based gating. The horizontal axis shows the weight $\alpha$ in Weighted RoPE and the vertical axis shows perplexity. We evaluate two settings for Sequence FE: $\tau > 0.4$ and $\tau < 0.2$. For Spectrum FE, we use $\tau < 0.6$ because outliers occur only at low values.

Table 1: Downstream task performance in original RoPE and WeightedRoPE.

| Model | HellaSwag | | TruthfulQA | | MMLU | |
|---|---|---|---|---|---|---|
| | baseline | +WeightedRoPE | baseline | +WeightedRoPE | baseline | +WeightedRoPE |
| Llama-4 17B | 66.67 | 66.67 | 97.32 | **97.99** | 60.05 | **60.81** |
| Llama-3 8B | 60.16 | 60.16 | 84.85 | 84.85 | 34.21 | 34.21 |
| Qwen3 8B | 58.94 | 58.92 | 95.31 | 95.31 | 57.89 | 57.89 |
| Gemma-2 9B | 61.02 | **61.21** | 98.83 | 98.83 | 72.81 | 72.81 |

**Results** To examine the contribution of outliers, we plotted the perplexity for each weight $\alpha$ to visualize how different weighting values affect model performance. Figure 4 presents the overall perplexity results. When SpectrumFE reduced the dimensions with $\tau < 0.2$, perplexity increased as the weight $\alpha$ decreased. This indicates that dimensions with $\tau < 0.2$ in SpectrumFE contribute to model performance and may be important components for the model—in other words, frequency bands may be important. Furthermore, when SpectrumFE reduced dimensions with $\tau > 0.4$, perplexity decreased slightly as the weight $\alpha$ became smaller. However, the overall performance remained nearly identical, suggesting that dimensions with outlier values of $\tau > 0.4$ may be unnecessary or redundant for the model. Next, when SequenceFE reduced dimensions with $\tau < 0.6$, perplexity decreased slightly as the weight $\alpha$ became smaller. Notably, the decrease in perplexity was larger for the Llama-4 model than for the other models, suggesting that periodicity may be unnecessary or redundant for the model. The Llama-3 model had a smaller impact on perplexity, but the trend was the same as other models.

## 5.3 DOWNSTREAM TASK

**Settings** Based on the above experimental results, we hypothesize that the outlier dimensions of SpectrumFE for $\tau > 0.4$ and the periodic dimensions of SequenceFE for $\tau < 0.6$ are redundant. To investigate whether these dimensions affect downstream tasks, we evaluated Weighted RoPE across multiple tasks. We fixed $\alpha$ to 0.1 for Weighted RoPE, applying the weight $\alpha$ to both the outlier dimensions of SpectrumFE at $\tau > 0.4$ and the periodic dimensions of SequenceFE at $\tau < 0.6$. Performance was measured on a diverse set of benchmark tasks including HellaSwag (Zellers et al., 2019), TruthfulQA (Lin et al., 2022), and MMLU (Hendrycks et al., 2021) to assess both reasoning and factual capabilities. [3]

**Results** Table 1 lists the experimental results. For all tasks, no significant difference was observed between RoPE, which performs no operations, and WeightedRoPE. Only the llama-4 model showed a slight improvement in performance. This suggests that the dimension where SpectrumFE becomes an outlier and the periodic dimension where SequenceFE decreases may not contribute to model performance and could be redundant.

---

[3]We additionally provide an evaluation of long-context generation on the Needle-in-a-Haystack task in Appendix E.

## 6 CONCLUSION

In this work, we introduced frequency entropy (FE), a metric that quantifies the effective utilization of each RoPE frequency dimension, and analyzed how the sinusoidal components of RoPE contribute to the model representation. SpectrumFE can identify the frequency band of RoPE, and reducing the contribution of this band degrades model performance, indicating it is a crucial component. Furthermore, SequenceFE can identify the periodic dimension of RoPE. Reducing the contribution of this dimension does not change model performance and may even improve it for some models. This suggests the periodic dimension may be redundant or unnecessary. Furthermore, these frequency bands and periodic dimensions do not exist in fixed dimensions. This suggests that some inconsistencies in prior work may stem from model-dependent frequency bands whose locations differ across heads and layers, rather than from absolute "low" or "high" frequency effects. Our framework provides a systematic method for interpreting such mixed features individually. Moreover, the fact that low-SequenceFE dimensions can be attenuated with minimal degradation indicates that these components reflect periodic signals induced by RoPE that the model does not functionally use. This finding highlights potential applications to RoPE-aware KV-cache compression and dimension pruning, and it offers a basis for future exploratory work on frequency-targeted architectural variants. We expect FE to function as a practical, model-independent diagnostic tool for positional encoding.

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

# A  ADDITIONAL EXPERIMENTS ON KEY

We also compute our spectral entropy (SpectrumFE and SequenceFE) from the keys in `Llama-4-Scout-17B-16E-Instruct` (Meta, 2025) model. The analysis procedure is the same as in Section 4.

**Analysis Results**   Figure 5 shows the scatter plots for each FE and the $\ell_2$-norm heatmap for the keys. In RoPE layers, we observe both frequency bands and periodic dimensions in queries and in keys. FE aligns with these observations and shows the same trend for keys. Keys contain a larger number of periodic dimensions than queries (e.g. 5-th Layer). In NoPE layers, we see the same qualitative pattern as in queries: periodic dimensions are rarer, while frequency bands are detected frequently.

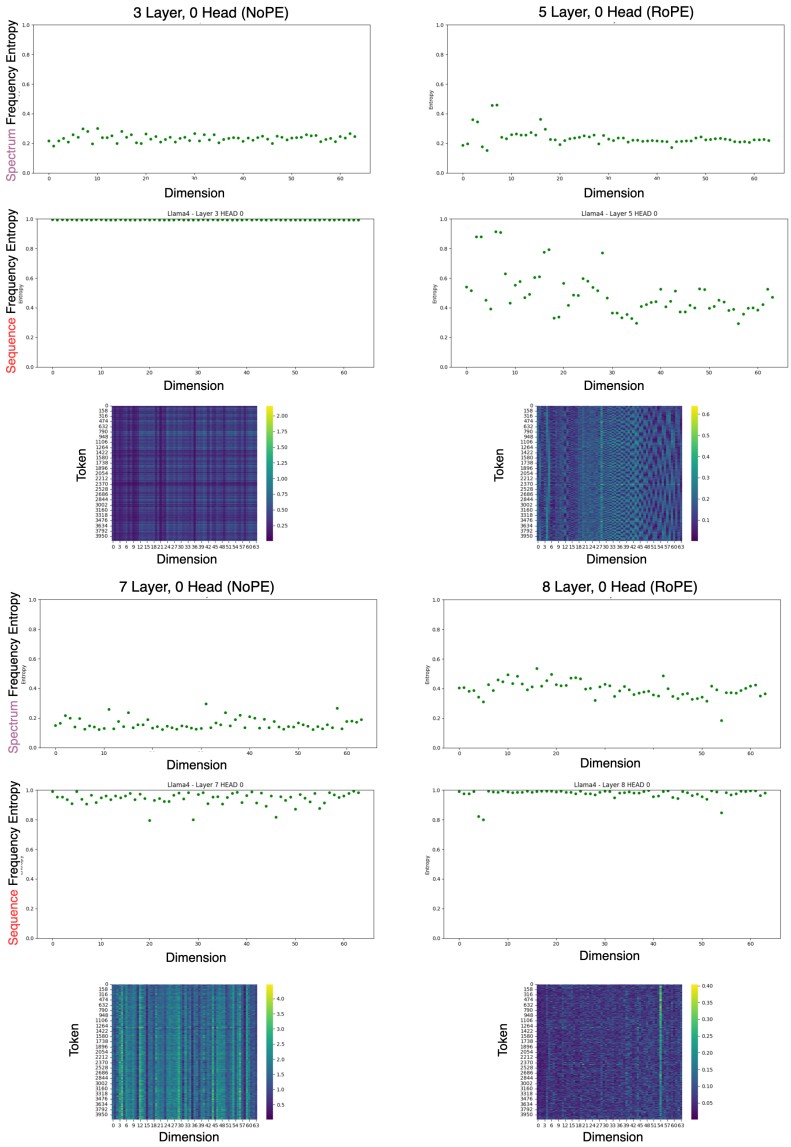

Figure 5: Scatter plots of each FE value in the `Llama-4-Scout-17B-16E-Instruct` model. Columns: layer 6, layer 1, layer 3 (left to right). Rows: SpectrumFE, SequenceFE, and key $\ell_2$-norm map. Top/middle: pair index in RoPE $j$ (x) vs. normalized entropy $\tilde{H}_j$ (y). Bottom: $j$ (y) vs. token index $n$ (x); color denotes $\|k_n^{(j)}\|_2$. All results are shown for head 0. The pair index $j$ is rotation patterns. Sequence length $L = 4096$.

## B   ADDITIONAL EXPERIMENTS ON ALL LAYERS

Figures 6 and 7 show the results of SpectrumFE and SequenceFE across all layers of the `Llama-4-Scout-17B-16E` model.

**SpectrumFE in all layers**   First, we discuss the scatter plots for SpectrumFE in Fig. 6. Comparing the shallow RoPE layer and the NoPE layer, the NoPE layer exhibits a significantly broader distribution, whereas the RoPE layer's distribution is less extensive. However, as the layer depth increases, the NoPE layer's distribution converges. Even in the RoPE layer, the distribution converges, though not as much as in the NoPE layer, suggesting the influence of deepening. On the other hand, low SpectrumFE values are observed regardless of layer depth, indicating that the frequency band is observed in every layer.

**SequenceFE in all layers**   Next, we discuss the scatter plots of SequenceFE in Fig. 7. Comparing the shallow RoPE layer and the NoPE layer, the RoPE layer exhibits a significantly broader distribution, while the NoPE layer's distribution is not as wide. This is exactly opposite to the trend observed in SpectrumFE. However, as the layer deepens, the NoPE layer's distribution widens, indicating that periodic dimensions temporarily emerge in the NoPE layer. Near the final layer, however, the periodic dimensions in the NoPE layer diminish. The RoPE layer maintains a certain number of periodic dimensions even at deeper layers. These periodic dimensions primarily exist near high frequencies. However, since these high-frequency periodic dimensions are not observed in the NoPE layer, NoPE may play a role in mitigating periodic dimensions.

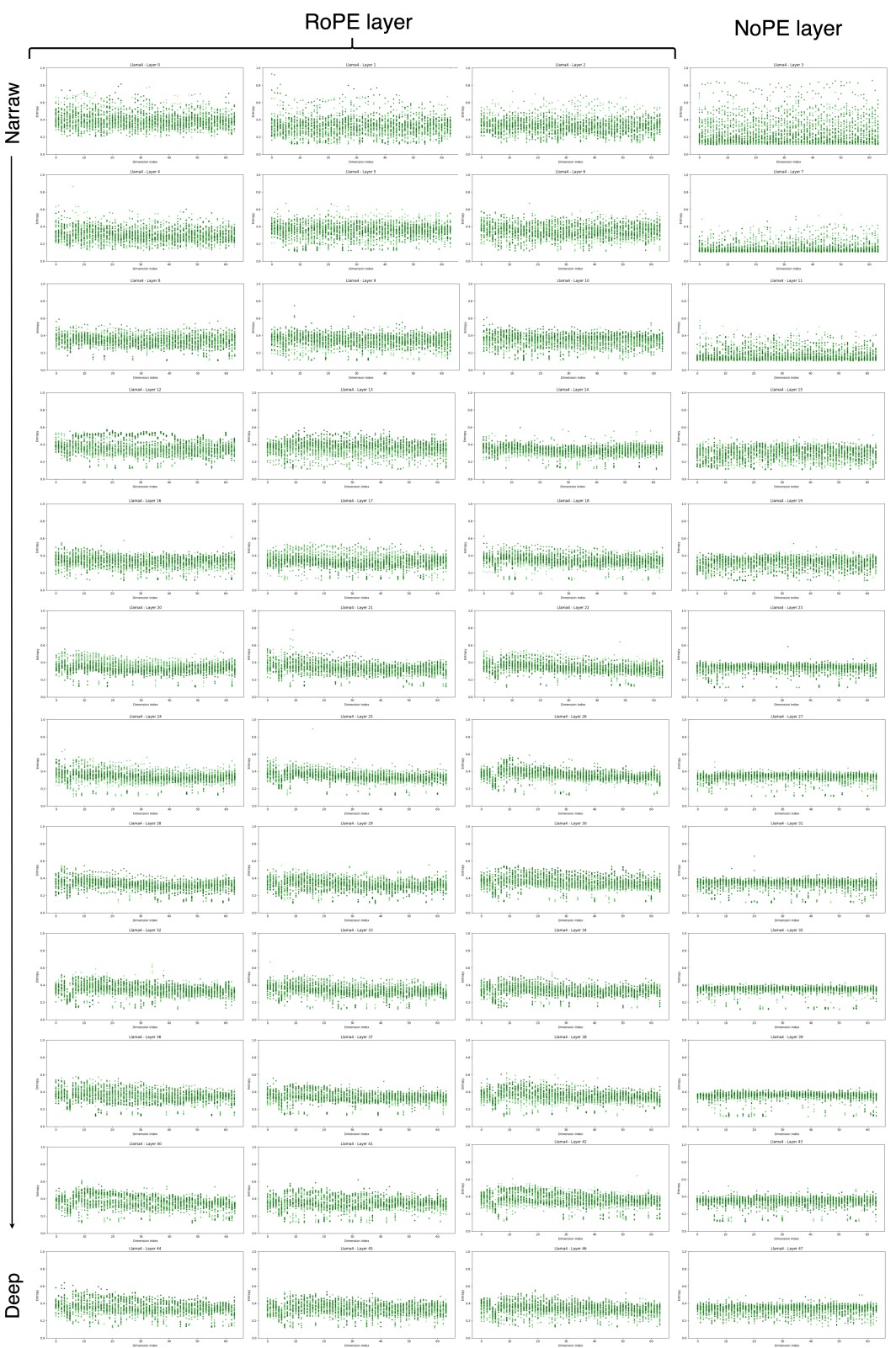

Figure 6: Layer-wise scatter plots of SpectrumFE across all attention heads in `Llama-4-Scout-17B-16E-Instruct`. The figure contains 48 panels arranged in 12 rows × 4 columns, with layer depth increasing left-to-right and then top-to-bottom (layers 0–47). Layers 3, 7, 11, 15, 19, 23, 27, 31, 35, 39, 43, and 47 use NoPE; all other layers use RoPE.

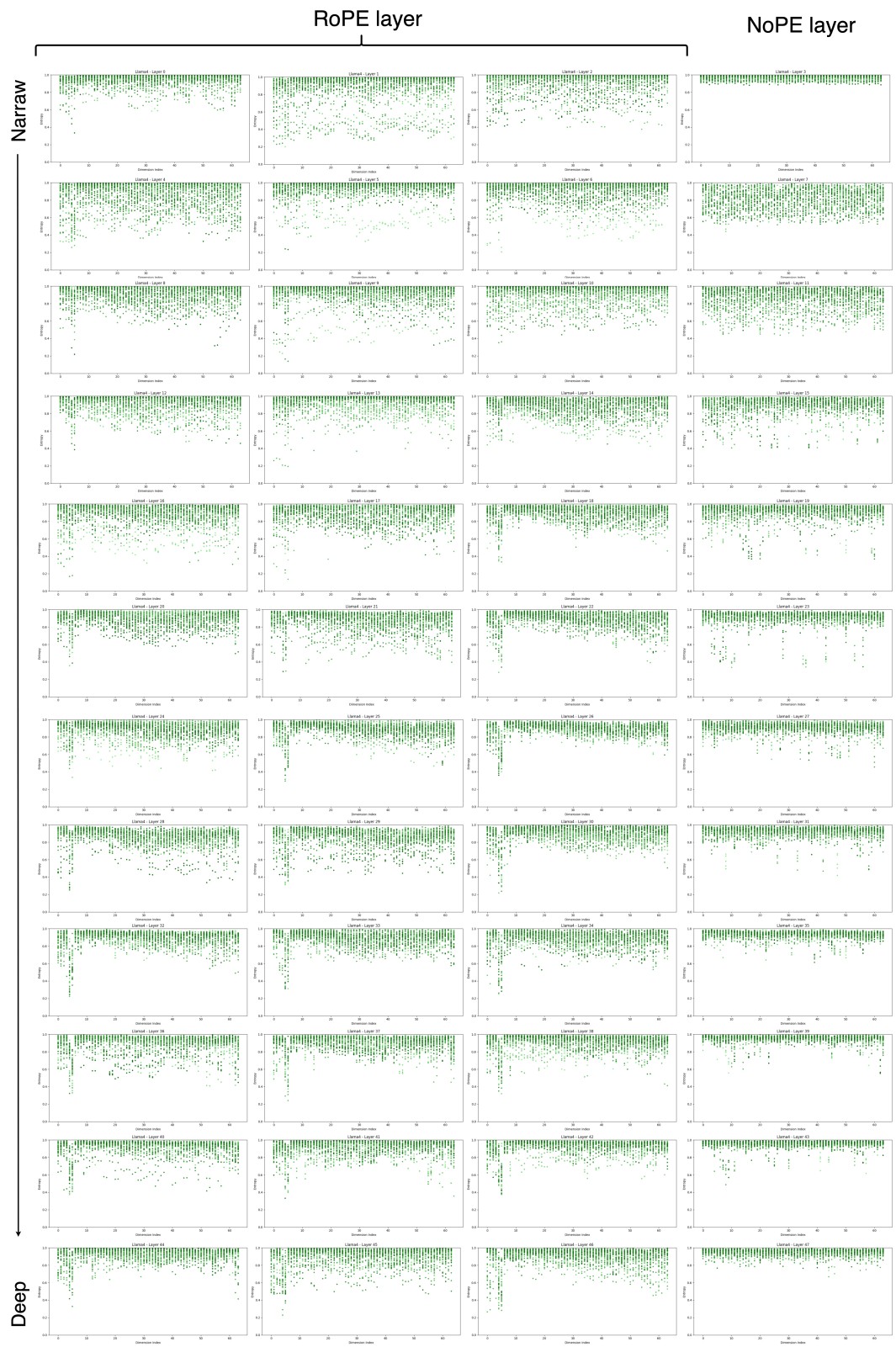

Figure 7: Layer-wise scatter plots of SequenceFE across all attention heads in Llama-4-Scout-17B-16E-Instruct. The figure contains 48 panels arranged in 12 rows × 4 columns, with layer depth increasing left-to-right and then top-to-bottom (layers 0–47). Layers 3, 7, 11, 15, 19, 23, 27, 31, 35, 39, 43, and 47 use NoPE; all other layers use RoPE.

## C    ADDITIONAL EXPERIMENTS ON OTHER ARCHITECTURES

We performed frequency entropy analysis not only on the `Llama-4-Scout-17B-16E` model but also on `Meta-Llama-3-8B`, `gemma-2-9b-it`, and `Qwen3-8B`. Note that unlike Llama-4, these models do not possess a NoPE layer. All experimental settings are the same as in Section 4.

**Llama-3**    Figure 8 shows the FE analysis for the Llama-3 model. Only selected salient features are marked. First, consistent with Llama-4 in Section 4, frequency bands are also observed in Llama-3. These bands appear in most heads. In contrast, no periodic dimensions are identified, and SequenceFE remains consistently high overall. The strongest band typically appears between the 39th and 42nd dimensions on average, which differs from the band locations in the RoPE layers of Llama-4.

**Gemma-2**    Figure 9 shows the FE analysis for the Gemma-2 model. Frequency bands are observed and they appear in most heads. In contrast, no periodic dimensions are identified, and SequenceFE remains consistently high overall. The strongest band typically appears between the 115th and 120th dimensions on average.

**Qwen-3**    Figure 10 shows the FE analysis for the Qwen-3 model. Frequency bands are observed and they appear in most heads. In contrast, no periodic dimensions are identified, and SequenceFE remains consistently high overall. The strongest band typically appears between the 48th and 50th dimensions on average.

Frequency bands appear across models yet their locations are model dependent. In Llama-3, Gemma-2, and Qwen-3, bands occur in many heads while the peak dimension differs across models, which suggests that band position is governed by the RoPE base, the training length, and architectural factors such as dimensionality and head configuration. In these model, periodic dimensions are ineffective or at most very limited, since SequenceFE remains consistently high and clear periodic components are not observed. Attenuating or pruning these periodic dimensions is likely to cause only a small drop in performance, which is consistent with the downstream task results in Section 5.3. Moreover, since the band is detected in most heads and explicitly reducing the contribution of the band dimensions lowers performance (as shown in Section 5.2), we conclude that a specific frequency range is commonly useful for attention computation.

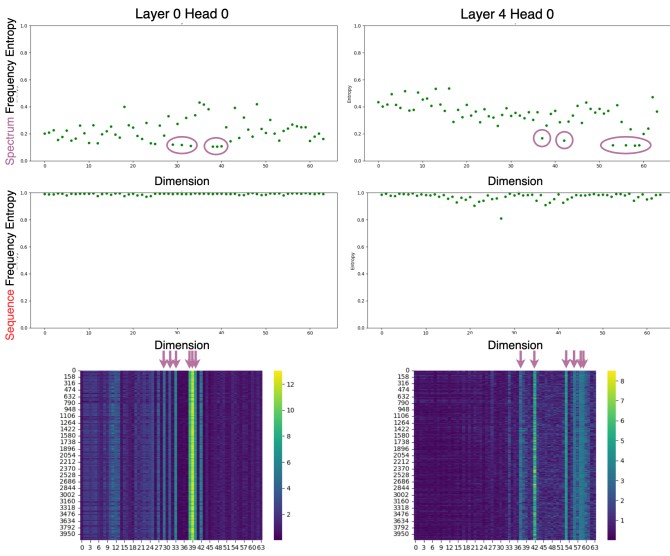

Figure 8: `Meta-Llama-3-8B` model (head 0). Columns: layer 0 and layer 4. Rows: SpectrumFE, SequenceFE, and query $\ell_2$-norm map. Top/middle: pair index $j$ (x) vs. normalized entropy $\tilde{H}_j$ (y). Bottom: $j$ (y) vs. token index $n$ (x); color denotes $\|q_n^{(j)}\|_2$. The pair index $j$ is rotation patterns. Sequence length $L = 4096$.

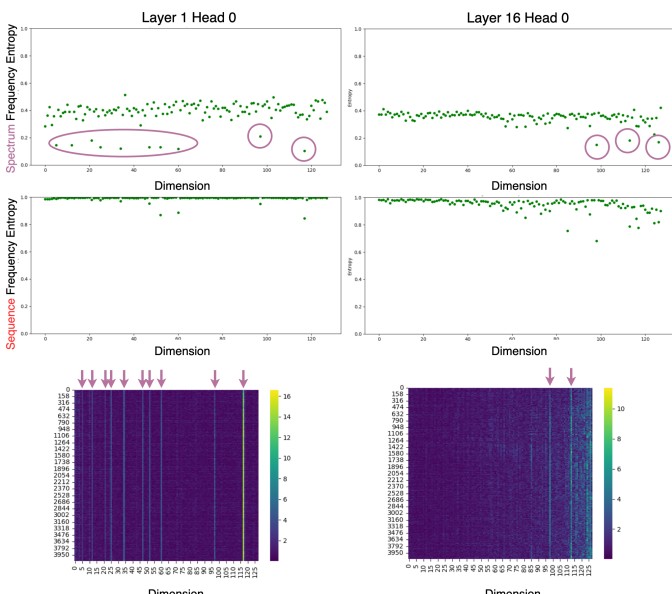

Figure 9: `Gemma-2-9b-it` model (head 0). Columns: layer 0 and layer 16. Rows: SpectrumFE, SequenceFE, and query $\ell_2$-norm map. Top/middle: pair index $j$ (x) vs. normalized entropy $\tilde{H}_j$ (y). Bottom: $j$ (y) vs. token index $n$ (x); color denotes $\|q_n^{(j)}\|_2$. The pair index $j$ is rotation patterns. Sequence length $L = 4096$.

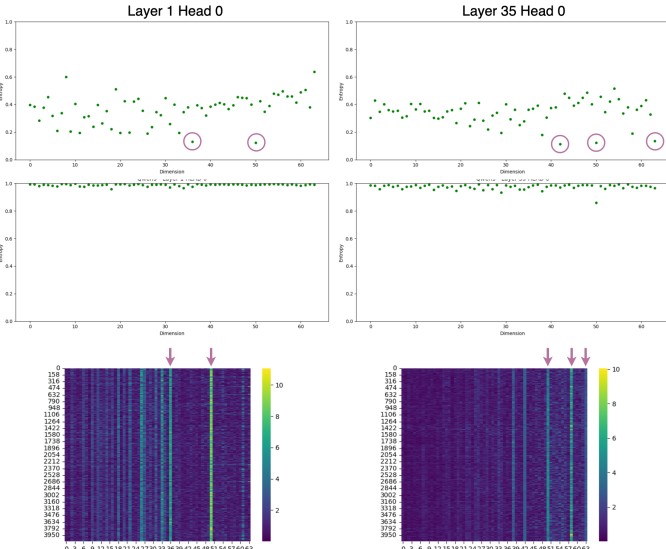

Figure 10: `Qwen-3-8B` model (head 0). Columns: layer 0 and layer 35. Rows: SpectrumFE, SequenceFE, and query $\ell_2$-norm map. Top/middle: pair index $j$ (x) vs. normalized entropy $\tilde{H}_j$ (y). Bottom: $j$ (y) vs. token index $n$ (x); color denotes $\|q_n^{(j)}\|_2$. The pair index $j$ is rotation patterns. Sequence length $L = 4096$.

## D  LIMITATION

While our work provides a unified analysis of RoPE's frequency usage, several limitations remain. Our intervention weighted RoPE does not fully disentangle whether performance preservation under SequenceFE-based attenuation reflects true redundancy or compensation from neighboring frequency components. A more granular causal test, such as single-pair perturbations, layer-specific interventions, or head-wise isolated ablations, would be required to conclusively rule out compensatory mechanisms. Finally, although we evaluate four architectures applying RoPE in all layers, our causal claims are limited to the scope of inference-only interventions. Our results should therefore be interpreted as providing first-order causal evidence rather than a complete causal theory of positional encoding. We regard the development of more precise, minimally confounded causal perturbations as an important direction for future work.

## E  ANALYZING LONG-CONTEXT BEHAVIOR THROUGH WEIGHTED ROPE

We evaluated Weighted RoPE using downstream benchmarks such as HellaSwag and MMLU in Section 5.3. These tasks primarily test general knowledge and reasoning, and therefore may not be sensitive to differences in positional encoding. To directly address this concern, we evaluated the impact of Weighted RoPE on long-context understanding in a setting explicitly designed to probe positional robustness. We used the Needle-in-a-Haystack (NIAH) task [4], which is widely used in studies of positional interpolation and long-context evaluation. Given the available GPU memory, we evaluate Llama-4 up to 33,564 tokens and Llama-3 up to 59,615 tokens, which correspond to the maximum feasible context lengths under our inference setup.

### E.1  EXPERIMENTAL SETUP

We evaluate Llama-3 and Llama-4 under our Weighted RoPE intervention in the NIAH setup. The Weighted RoPE settings are the same as in Section 5.3. We used the implementation of (Fu et al., 2024).

### E.2  RESULTS

The results for Llama-3 are shown in Figures 11 and 12, and the results for Llama-4 are shown in Figures 13 and 14.

Consistent with the downstream results reported in Table 1, Llama-3 showed no observable differences between the baseline RoPE and Weighted RoPE across all evaluated context lengths. The retrieval accuracy curves as well as the heatmap structure were effectively identical. This suggests that the RoPE dimensions identified by SpectrumFE as outliers or by SequenceFE as low-periodicity contribute little to Llama-3's long-context retrieval. In other words, the dimensions we suppress or down-scale appear truly redundant for this model, both in short-range downstream tasks and in explicit long-context settings.

For Llama-4, the overall retrieval pattern remained stable, but Weighted RoPE showed slight improvement around the 30k-token context. In the baseline model, this region appears as a cluster of red cells indicating retrieval failure; the Weighted RoPE produces milder failure or partial recovery.

These results provides two insights. (1) Long-context behavior can be affected by localized frequency-band adjustments.Unlike Llama-3, Llama-4 exhibits more sharply defined frequency-band structures in early layers. Weighted RoPE interacts with these bands, and selectively attenuating low-importance rotary pairs can mitigate localized positional instability. (2) Weighted RoPE does not harm long-context performance and may correct fragile regions.Reviewer concerns about potential degradation on long-context–specific tasks are therefore addressed: no regressions were observed, and in Llama-4 certain failure modes were slightly improved.

---

[4] https://github.com/gkamradt/LLMTest_NeedleInAHaystack

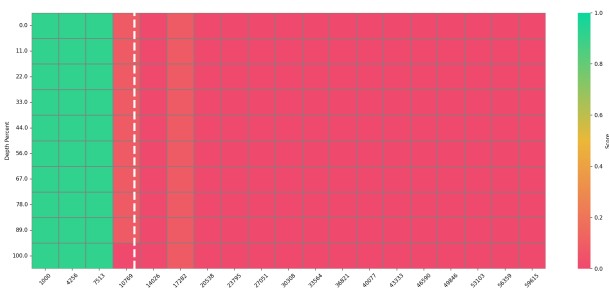

Figure 11: Results for NIAH on `Llama-3-8B`

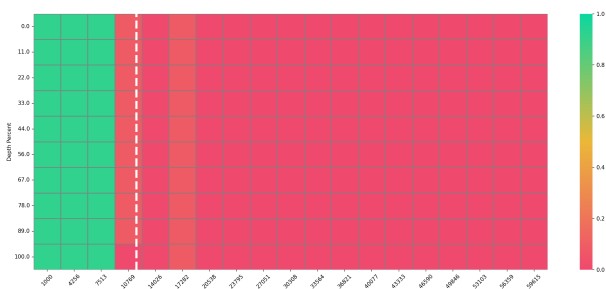

Figure 12: Results for NIAH task on `Llama-3-8B` with Weighted RoPE.

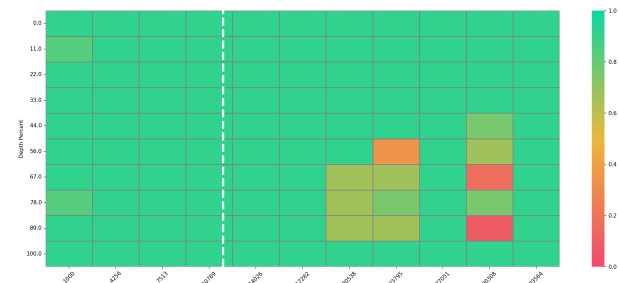

Figure 13: Results for NIAH task on `Llama-4-Scout-17B-16E-Instruct`.

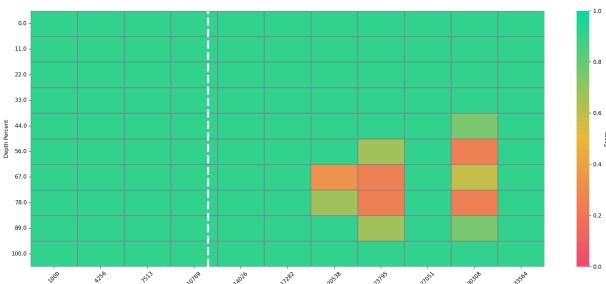

Figure 14: Results for NIAH task on `Llama-4-Scout-17B-16E-Instruct` with Weighted RoPE.

## F  ADDITIONAL EXPERIMENTS ACROSS DATASETS AND CONTEXT LENGTHS

We extend the analysis via frequency entropy in Section 4 on two additional datasets and across multiple context lengths. In addition to Wikitext-103, we conduct experiments on the C4 dataset (Raffel et al., 2020) [5], which consists of English text from Common Crawl [6], and the PG19 book corpus (Rae et al., 2020). We evaluate models at three context lengths: 2048, 4096, and 8192 tokens.

### F.1  ANALYSIS ACROSS CONTEXT LENGTHS

The results for Llama-4 using Wikitext-103 are shown in Figure 15. When we vary the context length, we observe slightly differences in SpectrumFE, SequenceFE, and the 2-norm maps. These changes are expected because the input text itself differs across context-length settings. However, the overall distribution of both SpectrumFE and SequenceFE remains largely unchanged. The frequency bands captured by SpectrumFE persist across all context lengths, and their locations remain stable. Similarly, the periodic patterns captured by SequenceFE also remain present at every context length, with their positions unchanged. As shown in the left 2-norm maps of Figure 15, the visual appearance of the periodic patterns changes as the context length varies, but the underlying dimensions where these patterns occur stay the same. Taken together, these findings indicate that both SpectrumFE and SequenceFE consistently can capture their respective structural properties, and these properties do not shift when the context length is changed.

### F.2  ANALYSIS ACROSS DATASETS

Figure 16 and Figure 17 show the Llama-4 results on C4 and PG19 respectively, evaluated at context lengths of 2048, 4096, and 8192 tokens. We observe no major differences across context lengths, and changing the dataset similarly leaves the overall distributions of SpectrumFE and SequenceFE largely unchanged. The frequency bands captured by SpectrumFE persist in both the C4 and PG19 datasets, and their locations remain stable. Similarly, the periodic patterns captured by SequenceFE continue to appear in both datasets, with their positions unchanged. Taken together, these findings indicate that both SpectrumFE and SequenceFE consistently can capture their respective structural properties, and these properties do not shift when the input sequence is changed.

---

[5]We use the 'en' validation split from the processed version provided at the following URL: https://huggingface.co/datasets/allenai/c4

[6]https://commoncrawl.org/

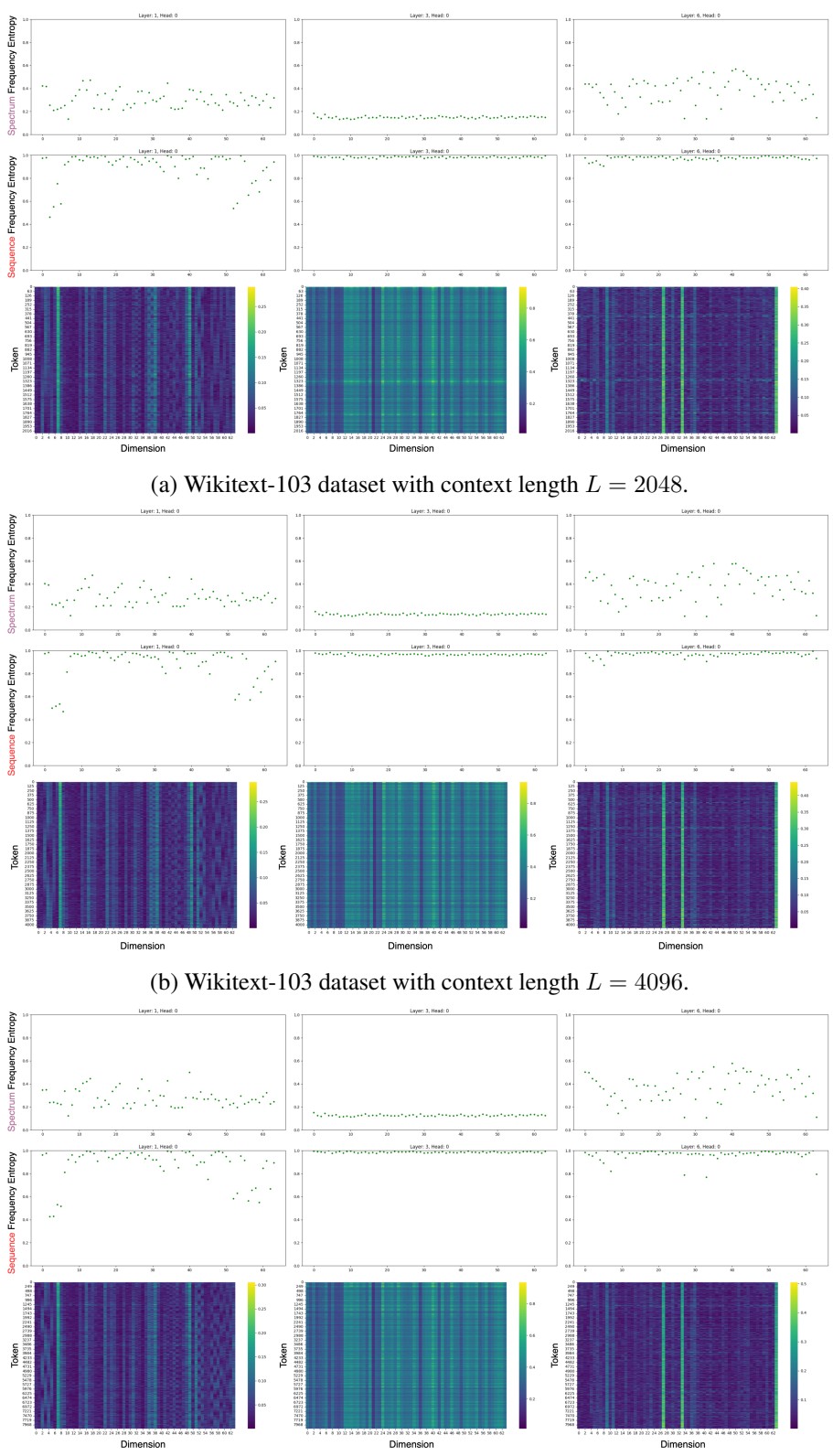

(a) Wikitext-103 dataset with context length $L = 2048$.

(b) Wikitext-103 dataset with context length $L = 4096$.

(c) Wikitext-103 dataset with context length $L = 8192$.

Figure 15: Scatter plots of each FE value in the `Llama-4-Scout-17B-16E-Instruct` model. Columns: layer 6, layer 1, layer 3 (left to right). Rows: SpectrumFE, SequenceFE, and query $\ell_2$-norm map. All results are shown for head 0.

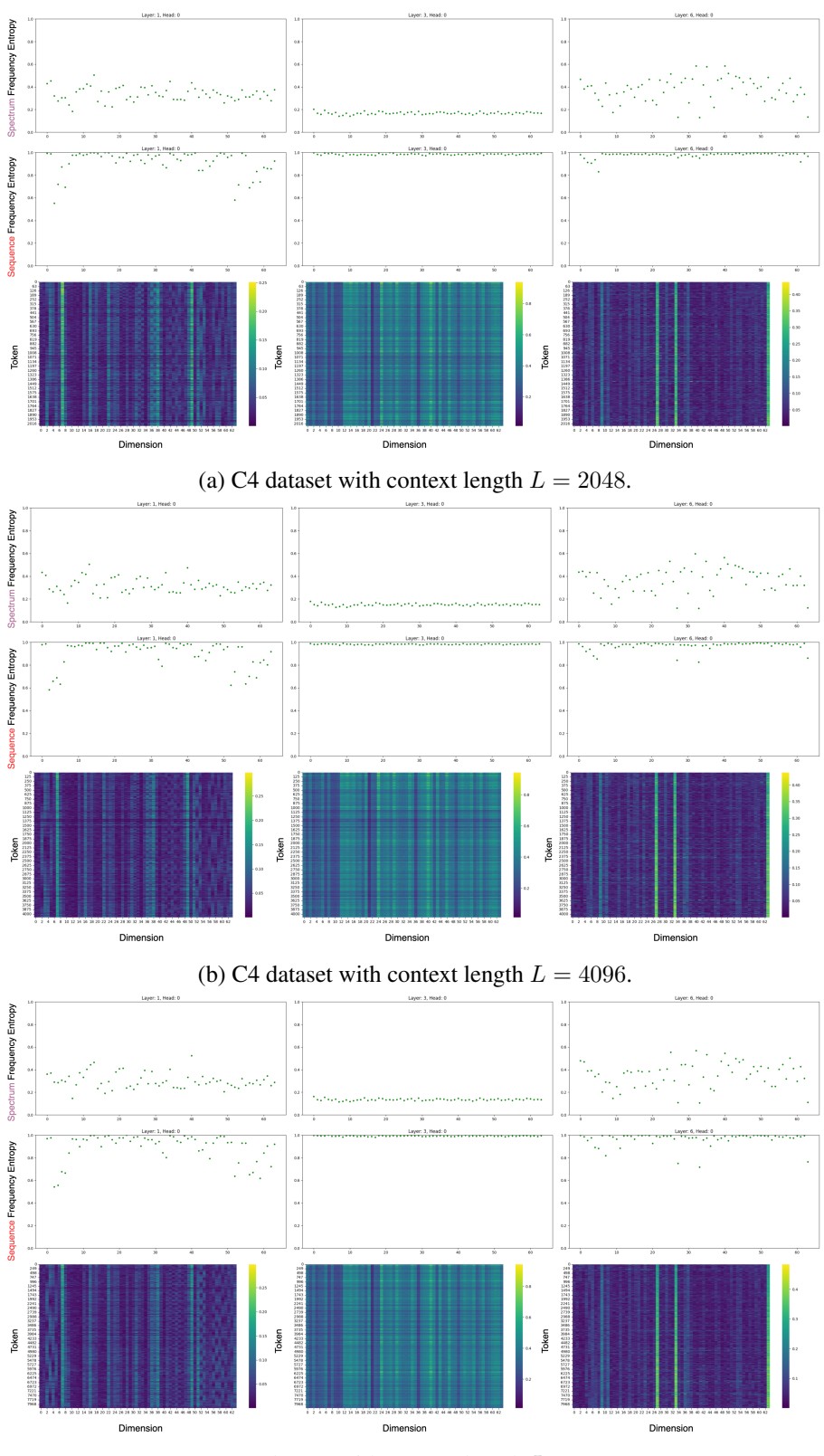

(a) C4 dataset with context length $L = 2048$.

(b) C4 dataset with context length $L = 4096$.

(c) C4 dataset with context length $L = 8192$.

Figure 16: Scatter plots of each FE value in the `Llama-4-Scout-17B-16E-Instruct` model. Columns: layer 6, layer 1, layer 3 (left to right). Rows: SpectrumFE, SequenceFE, and query $\ell_2$-norm map. All results are shown for head 0.

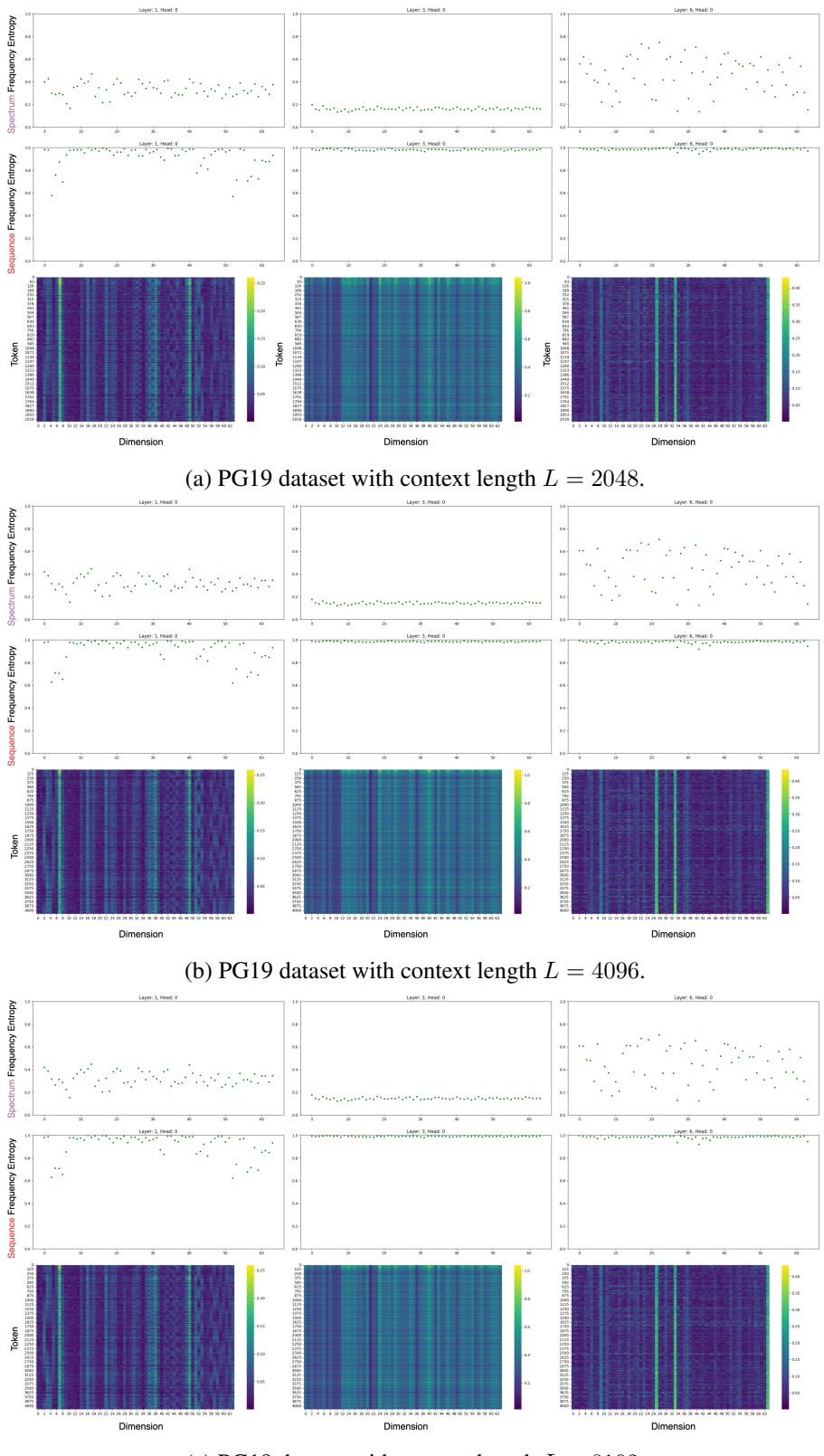

(a) PG19 dataset with context length $L = 2048$.

(b) PG19 dataset with context length $L = 4096$.

(c) PG19 dataset with context length $L = 8192$.

Figure 17: Scatter plots of each FE value in the `Llama-4-Scout-17B-16E-Instruct` model. Columns: layer 6, layer 1, layer 3 (left to right). Rows: SpectrumFE, SequenceFE, and query $\ell_2$-norm map. All results are shown for head 0.

## G  FREQUENCY ENTROPY COMPARISON BETWEEN ROPE-ONLY AND NOPE-ONLY MODELS

Llama-4 interleaves RoPE and NoPE layers, causing the effects of the two positional schemes to be mixed within the same network. To disentangle their individual contributions, we pretrain two small models from scratch: one that uses only RoPE and another that uses only NoPE. We then evaluate their layer-wise Frequency Entropy (FE) to examine the independent behavior induced by each encoding.

### G.1  EXPERIMENTAL SETUP

For pre-training from scratch, we perform a comparative evaluation with a Transformer-based language model (Baevski & Auli, 2019). The dimensionality of the word embedding $d_{model}$ is 1024, the number of heads $N$ is 8, the dimensionality of the heads $d$ is 128, and the number of layers is 16. This implementation used the fairseq (Ott et al., 2019)-based code provided in a previous work(Press et al., 2022), and all hyperparameters were set to the same values as those in the literature(Press et al., 2022). We use the Nesterov's Accelerated Gradient (NAG) optimizer with momentum 0.99, following the Fairseq nag implementation in (Ott et al., 2019). The learning rate schedule is a cosine scheduler: it is initialized at 1e-7, linearly warmed up to 1.0 during the first 16,000 updates, and then decayed with a cosine schedule down to 1e-4. The value 9216 denotes the maximum number of tokens per batch per GPU, with a sequence length of 512 tokens. We train for 286,000 updates, which corresponds to approximately 205 epochs on WikiText-103.  We used the WikiText-103 dataset (Merity et al., 2017), which consists of over 103 million tokens of English Wikipedia articles. This setup used in (Press et al., 2022; Oka et al., 2025) [7], where the goal is to analyze structural differences in positional encodings.

### G.2  RESULTS

Figure 18 reports the FE measurements for layer 0 of the RoPE-only model, and Figure 19 reports the results for its final layer, layer 15. Figures 20 and 21 provide the corresponding FE results for the NoPE-only model.

First, in the RoPE-only model, the SpectrumFE results in the top row reveal clear frequency bands. The locations of these bands vary across heads and layers, but the band structure itself is consistently present in every head and layer we examined. This behavior matches what we observed in the RoPE layers of Llama-4, as well as in Llama-3, Gemma, and Qwen. In contrast, SequenceFE remains high across all dimensions, indicating little periodic structure. This differs from Llama-4 but aligns with the RoPE-only behavior reported for Llama-3, Gemma, and Qwen in Appendix C.

Next, in the NoPE-only model, the SpectrumFE results show that frequency bands do not appear in all heads. Overall SpectrumFE values are lower than in the RoPE-only model, and the number of dimensions forming identifiable bands is also smaller. Some heads show no band structure at all. For example, the right-side in Figure 18 (0 Layer, 7 Head) displays a pattern closer to noise in its 2-norm map. Moreover, in the final layer, the frequency bands disappear entirely. SequenceFE remains high throughout, showing no periodic patterns. This is expected because NoPE does not introduce any periodic information.

These observations suggest that a NoPE-only model tends to weaken or wash out frequency bands as depth increases. RoPE, in contrast, consistently maintains band structure even in deeper layers. Finally, in architectures like Llama-4 that interleave RoPE and NoPE layers, NoPE may reinforce the band structure introduced by RoPE while also amplifying the periodic components associated with RoPE. This interaction between the two positional schemes may explain the distinct FE patterns observed in mixed-layer settings.

---

[7] https://github.com/ofirpress/attention_with_linear_biases

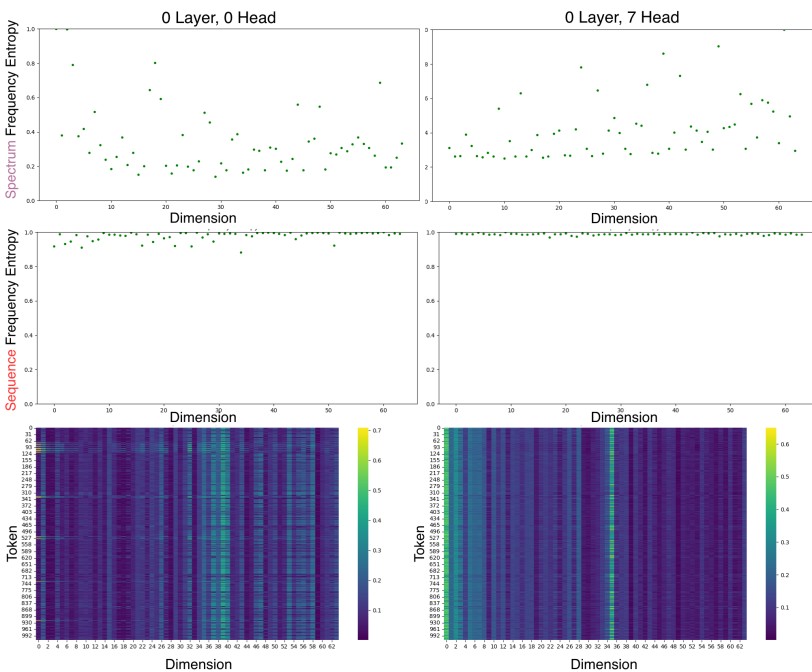

Figure 18: Frequency Entropy (FE) scatter plots for a **RoPE-only** model pretrained from scratch. Shown are the 0th and 7th heads of **layer 0**. Rows depict SpectrumFE, SequenceFE, and the query $\ell_2$-norm map.

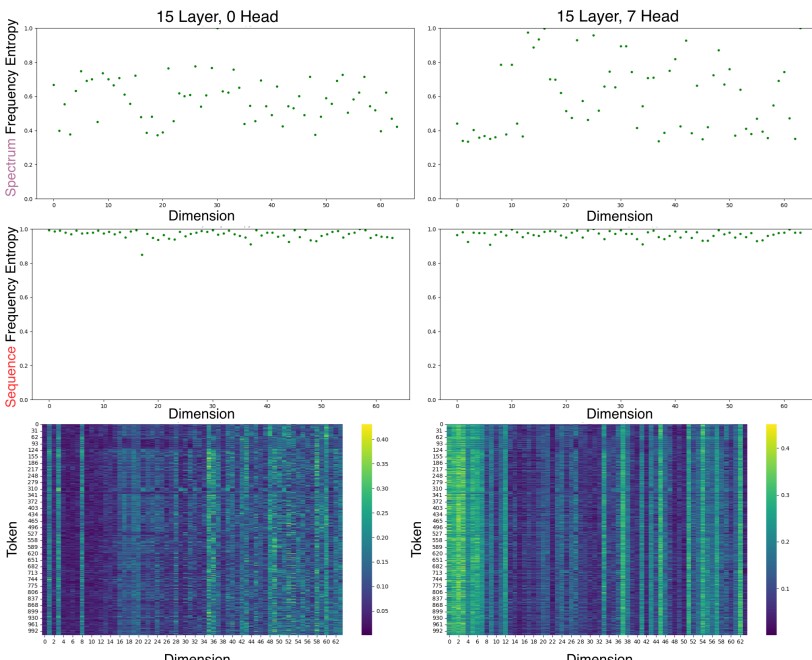

Figure 19: Frequency Entropy (FE) scatter plots for a **RoPE-only** model pretrained from scratch. Shown are the 0th and 7th heads of **last layer**. Rows depict SpectrumFE, SequenceFE, and the query $\ell_2$-norm map.

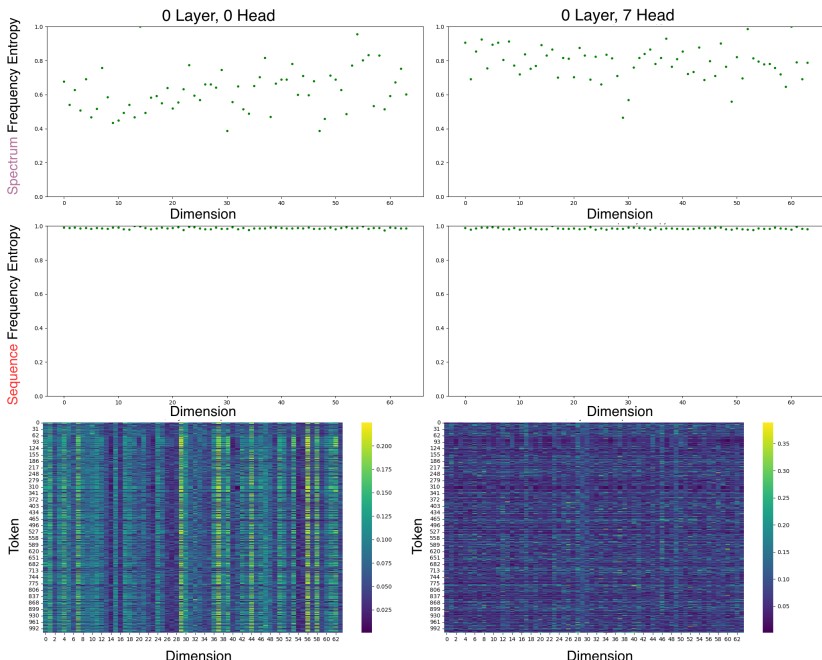

Figure 20: Frequency Entropy (FE) scatter plots for a **NoPE-only** model pretrained from scratch. Shown are the 0th and 7th heads of **layer 0**. Rows depict SpectrumFE, SequenceFE, and the query $\ell_2$-norm map.

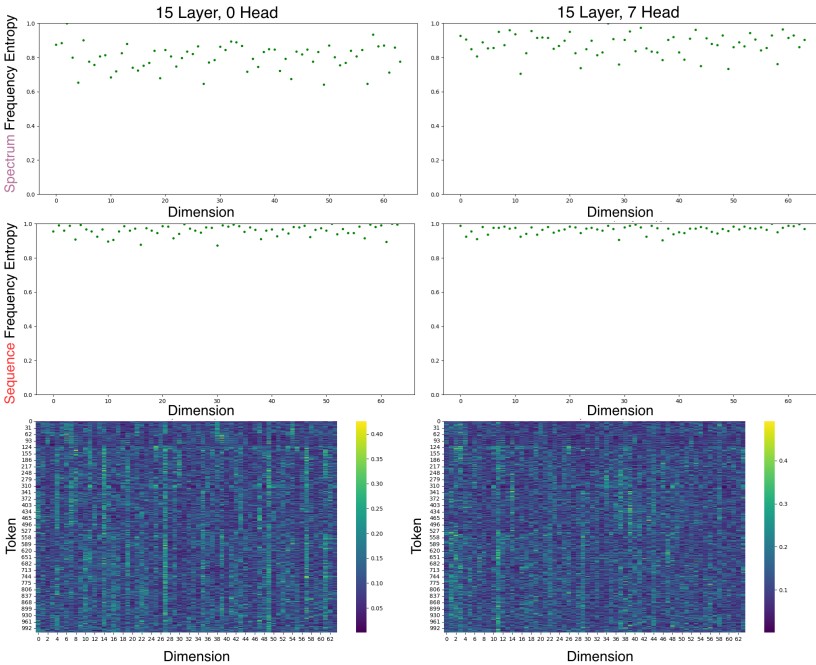

Figure 21: Frequency Entropy (FE) scatter plots for a **NoPE-only** model pretrained from scratch. Shown are the 0th and 7th heads of **last layer**. Rows depict SpectrumFE, SequenceFE, and the query $\ell_2$-norm map.

