# OpenReview forum: "Probing Rotary Position Embeddings through Frequency Entropy"
_ICLR.cc/2026/Conference — ICLR 2026 Poster_

### Official Review · Reviewer_672v · 2025-10-22

**Soundness:** 3
**Presentation:** 3
**Contribution:** 1
**Rating:** 4
**Confidence:** 3

**Summary:**

1. SpectrumFE aligns with distinct band-limited patterns in l2-norm maps.
2. SequenceFE is sensitive to periodic structures along the token axis.
3. From SpectrumFE, the shallow NoPE layer (layer 3) exhibits more frequency bands than the corresponding RoPE layer. Conversely, from SequenceFE, the NoPE layer lacks rotating pairs, indicating an absence of clear periodic oscillations.
4. NoPE may therefore suppress the periodic structures characteristic of RoPE while amplifying specific frequency bands.
5. Dimensions with τ < 0.2 in SpectrumFE appear to contribute positively to model performance, suggesting that these frequency bands are important functional components.
6. Dimensions with τ > 0.4 may be redundant or unnecessary for model performance.
7. Together, these results suggest that dimensions where SpectrumFE becomes an outlier and where SequenceFE decreases may not contribute meaningfully to performance and could be removed without loss.

**Strengths:**

* The authors introduce two novel metrics, SpectrumFE and SequenceFE, to analyze the frequency characteristics of RoPE embeddings.
* They identify a variety of interesting phenomena associated with these metrics.
* Ablation studies are performed to assess the effects of dampening or removing specific frequency bands.
* The paper attempts to reconcile previously conflicting findings regarding frequency importance in RoPE.

**Weaknesses:**

* The practical impact of the findings appears limited. The only direct link to model performance is provided by the ablation study (Section 5.3), which shows that removing dimensions where SpectrumFE becomes an outlier or where SequenceFE decreases does not degrade performance. This limits the applicability and relevance of the results for researchers and practitioners.
* The authors’ explanation of prior conflicting results is anecdotal rather than empirical. No targeted experiments are provided to demonstrate that earlier discrepancies arose from confounding factors. The claim that their findings “resolve the confusion of previous research” lacks sufficient supporting evidence.
* It remains unclear what SpectrumFE and SequenceFE truly measure. The paper states only that SpectrumFE aligns with band-limited l2 patterns and SequenceFE captures periodicity along the token axis-insufficient for understanding their mechanistic or theoretical grounding.
* Overall, the study focuses primarily on how these metrics interact with the model, rather than revealing new, interpretable mechanisms that explain how RoPE or LLMs function at a deeper level.

**Questions:**

1. Could the authors provide greater intuition for what SpectrumFE and SequenceFE actually measure? How do these metrics relate to underlying model mechanics? Examples or analogies could greatly enhance interpretability.
2. How might these findings be useful for researchers and practitioners? Could the authors elaborate on how the results should inform future research directions or guide the design of LLM architectures?
3. The author write: "We expect FE to function as a practical, model-independent diagnostic tool for position coding." Could the authors expand on this? What decisons or actions would a practitioner take based on FE measurements?

---

> ### Author Response · Authors · 2025-11-22
> **Author Response (1/3)**
>
> Thank you very much for reviewing our paper. We sincerely appreciate your valuable feedback. Please find our responses below. All revisions made to the manuscript are highlighted in red.
>
> > **(Weakness 1) The practical impact of the findings appears limited. The only direct link to model performance is provided by the ablation study (Section 5.3), which shows that removing dimensions where SpectrumFE becomes an outlier or where SequenceFE decreases does not degrade performance. This limits the applicability and relevance of the results for researchers and practitioners.**
>
> Thank you for raising this important point. Our goal in this work is to develop a general analytical framework for understanding how RoPE distributes and utilizes frequency components across layers and heads. While we do not aim to introduce a new positional encoding or demonstrate direct performance improvements, the framework offers two concrete forms of practical value.
>
> (1) **Revealing structural regularities that prior analyses could not capture.** Most prior work evaluated RoPE along a 1-D “low vs. high frequency” axis, which naturally led to conflicting conclusions. And this high-frequency/low-frequency analysis cannot be applied to schemes like NoPE that do not use position coding. By separating SpectrumFE (how energy is allocated across rotary pairs) and SequenceFE (how RoPE induces per-token periodicity), we uncover that RoPE or NoPE has band-structured, layer-dependent, and head-specific. This enables reasoning about which frequency regions are actually exploited during attention, something that was not accessible through existing analyses.
>
> (2) **Identifying systematic redundancy that directly informs compression and design.**
> The fact that low-SequenceFE dimensions can be attenuated with minimal degradation indicates that these components reflect periodic signals induced by RoPE that the model does not functionally use. This finding highlights potential applications to RoPE-aware KV-cache compression and dimension pruning,
> and it offers a basis for future exploratory work on frequency-targeted architectural variants.
>
> Thus, even though the present work focuses on analysis rather than proposing a new encoding, FE provides a practical diagnostic tool that clarifies which parts of the positional spectrum matter, why prior observations conflicted, and how future designs can exploit band-structured usage.
>
> **We have clarified this scope and its implications in the revision. (Section 6 Conclusion)**
>
> > **(Weakness 2)The authors’ explanation of prior conflicting results is anecdotal rather than empirical. No targeted experiments are provided to demonstrate that earlier discrepancies arose from confounding factors. The claim that their findings “resolve the confusion of previous research” lacks sufficient supporting evidence.**
>
> Thank you for pointing this out.
> Our claim is instead that the *patterns we observe empirically* provide a natural explanation for why such contradictions arose.
> Across all four models we evaluate, we observe two consistent empirical phenomena:
>
> (1) **RoPE usage forms frequency bands whose locations shift across layers and heads.**
> This directly implies that the “importance” of a given rotary pair depends on where the band lies in that specific layer/head—something prior analyses, which evaluated RoPE only along a global “low vs. high frequency” axis, could not account for.
>
> (2) **SpectrumFE and SequenceFE separate two distinct effects** which standard metrics mix together.
>
> - Structural band allocation across rotary pairs
>
> - Token-level periodicity induced by RoPE’s rotational phase
>
> **We have revised the description of Section 6 (Conclusion) you pointed out as follows.**
> >  This suggests that some inconsistencies in prior work may stem from model-dependent frequency bands whose locations differ across heads and layers, rather than from absolute “low” or “high” frequency effects. Our framework provides a systematic method for interpreting such mixed features individually. (L487-490)

---

> ### Author Response · Authors · 2025-11-22
> **Author Response (2/3)**
>
> > **(Weakness 3) It remains unclear what SpectrumFE and SequenceFE truly measure. The paper states only that SpectrumFE aligns with band-limited l2 patterns and SequenceFE captures periodicity along the token axis-insufficient for understanding their mechanistic or theoretical grounding.**
>
> Thank you for pointing this out. We clarify the conceptual roles:
>
> - **SpectrumFE**Measures *how narrowly concentrated* the local frequency content (via STFT) is.Low SpectrumFE indicates strong **frequency-band structure**, meaning the model consistently allocates energy to specific rotary pairs.
> - **SequenceFE**Measures *global periodicity* of the RoPE-transformed signal (via DFT).Low SequenceFE indicates near–single-tone oscillation driven by RoPE’s fixed rotational phase, rather than content.These are the “mechanically induced periodic pairs.”
>
> Together, the two entropies separate:
>
> - structural band allocation across pairs
>
> - per-token periodicity induced by RoPE itself
>
> These features are otherwise entangled in standard analyses.
> **We had already explained the mechanistic and theoretical basis of Frequency Entropy in Section 4.3, but we have added further details. (L367-381)**
>
> > **(Weakness 4) Overall, the study focuses primarily on how these metrics interact with the model, rather than revealing new, interpretable mechanisms that explain how RoPE or LLMs function at a deeper level.**
>
> Thank you for your feedback. This study does not aim to establish new fundamental functional mechanisms for RoPE or LLMs.
> Our objective is to provide a unified *analysis* of the frequency structures induced by RoPE and demonstrate how these structures consistently emerge across layers, heads, and architectures.
> Explaining how LLMs function at deeper levels is intriguing but falls outside our scope.

---

> ### Author Response · Authors · 2025-11-22
> **Author Response (3/3)**
>
> > **(Questions 1) Could the authors provide greater intuition for what SpectrumFE and SequenceFE actually measure? How do these metrics relate to underlying model mechanics? Examples or analogies could greatly enhance interpretability.**
>
> Thank you for your question.  Please see reply for Weakness 3.
>
> > **(Questions 2) How might these findings be useful for researchers and practitioners? Could the authors elaborate on how the results should inform future research directions or guide the design of LLM architectures?**
>
> Thank you for your question.
> Our findings provide practical guidance for both researchers and practitioners, especially as modern LLMs increasingly adopt hybrid positional encodings (e.g., combinations of RoPE and NoPE as in Llama-4).
> Previous analyses relied on a “low- vs. high-frequency” view of RoPE, which cannot be applied to NoPE because it does not generate frequency components. **In contrast, FE offers a unified framework that can analyze both RoPE and NoPE under the same lens, enabling systematic comparisons when designing or evaluating new positional schemes.**
>
> To understand how each component behaves in isolation, we additionally pre-trained models using RoPE-only and NoPE-only configurations (details in Appendix G). The key structural differences were:
>
> - NoPE-only models
>     - exhibit thin bands only in shallow layers;
>     - lose band structure entirely in deeper layers;
>     - show no periodicity.
> - RoPE-only models
>     - maintain band structure across all layers and heads;
>     - show limited periodicity (no strong periodic components).
>
> When combined, as in Llama-4, **NoPE strengthens the frequency band induced by RoPE**, but **it also amplifies RoPE’s periodic components**, creating a clear **tradeoff** between beneficial band enhancement and undesirable periodicity amplification.
> **This tradeoff has direct implications for model design.**
>
> FE can quantify how different mixtures of RoPE and NoPE shift (i) band retention, (ii) periodicity, and (iii) layer-wise stability, allowing practitioners to:
>
> 1. choose optimal mixing ratios when designing hybrid encodings;
> 2. diagnose when NoPE suppresses or enhances RoPE’s frequency structures;
> 3. identify configurations that preserve band structure without introducing strong periodic artifacts.
>
> In this sense, FE functions as a practical diagnostic tool for exploring, evaluating, and tuning positional encoding choices in future LLM architectures.
>
> > **(Questions 3) The author write: "We expect FE to function as a practical, model-independent diagnostic tool for position coding." Could the authors expand on this? What decisons or actions would a practitioner take based on FE measurements?**
>
> Our FE will supports several actionable decisions:
>
> - Model compression: Remove or down-weight rotary pairs with low SequenceFE.
> - Architecture tuning: Identify important frequency bands via SpectrumFE before position interpolation θ or designing alternative positional encodings.
> - Training diagnostics: Monitor FE evolution to detect under-utilization of specific RoPE bands.
>
> Thus FE is not merely descriptive, it is a model-independent, scale-free diagnostic that guides compression, design.
>
> ---
>
> Thank you for the thoughtful and practically oriented feedback. It has been very helpful in improving the clarity and applicability of our work.  If you have any further questions or requests, we would be more than happy to address them.

---

> > ### Comment · Reviewer_672v · 2025-11-24
> >
> > Thank you for the detailed rebuttal.
> >
> > I acknowledge that SpectrumFE and SequenceFE have the potential to become interesting analytical tools for examining model dynamics during training. However, in the absence of experiments that specifically demonstrate how these measures can effectively guide architectural design decisions or support training diagnostics, it remains difficult to recommend this work for publication.
> >
> > For these reasons, I am maintaining my original evaluation.
> >
> > I wish the authors the best in thei research.

---

> ### Author Response · Authors · 2025-11-27
>
> Thank you very much for your thoughtful follow-up comment. We also sincerely appreciate your positive remark that SpectrumFE and SequenceFE may serve as useful analytical tools.
>
> We would like to clarify our intended scope to avoid misunderstanding.
> **For the two points you mentioned (architecture tuning and training diagnostics) our intention was to outline potential future applications, not to claim that these were validated contributions of this submission. These items were not part of the original claims; they were added only to clarify the possible scope of FE after your earlier questions regarding practical applications (your Question 3).** To prevent overinterpretation, we have revised the Conclusion section (L492–493, blue text).
>
> We sincerely appreciate your feedback, which helped us refine the scope and avoid unintended overstatements. Thank you again for your careful evaluation.

---

### Official Review · Reviewer_dKd1 · 2025-10-30

**Soundness:** 3
**Presentation:** 4
**Contribution:** 3
**Rating:** 8
**Confidence:** 3

**Summary:**

This paper introduces Frequency Entropy (FE), a novel metric to analyze how different frequency dimensions in Rotary Position Embeddings (RoPE) are utilized in Transformers. Through systematic analysis of models like Llama-4, the authors show that FE identifies two key structures: frequency bands (via SpectrumFE) and periodic dimensions (via SequenceFE). They find that while frequency bands are essential for model performance, periodic dimensions are often redundant and can be attenuated without harming downstream task accuracy, offering a unified explanation for previously conflicting findings and providing a practical diagnostic tool for positional encoding design.

**Strengths:**

1. Novel Metric and Framework. Introduces Frequency Entropy (FE)—a new quantitative metric to analyze RoPE's frequency-wise utilization, unifying previously conflicting observations about high- vs. low-frequency roles. The dual metrics (SpectrumFE and SequenceFE) offer a systematic, model-agnostic diagnostic tool.
2. Rigorous and Multi-Model Validation.Strong empirical foundation with experiments across multiple architectures (Llama-4, Llama-3, Gemma-2, Qwen-3), probing both queries and keys. Combines entropy analysis with intervention (Weighted RoPE) to validate functional importance of frequency bands and periodicity.
3. Well-Structured and Accessible. Clear problem framing, method description, and visualizations (entropy scatter plots, norm heatmaps) make complex frequency dynamics interpretable. Appendices extend analysis to keys and all layers, enhancing reproducibility.
4. Resolves Prior Conflicts and Informs Design. Resolves contradictions in prior work by showing frequency bands (low SpectrumFE) are essential, while periodic dimensions (low SequenceFE) are often redundant. Offers practical implications for model efficiency and positional encoding design.

**Weaknesses:**

1. Experiments are conducted primarily on the Wikitext-103 dataset. To fully support the claim that FE is a "general diagnostic," the analysis should include diverse text datasets to verify whether the findings generalize beyond specific tasks.
2. The paper establishes correlations between FE values and model behaviors but does not rigorously prove causality. For instance, while attenuating low-SequenceFE dimensions doesn't hurt performance, it remains unclear if this is because they are truly redundant or if the model compensates via other mechanisms. A more controlled ablation would strengthen the causal claim.
3. The paper studies Llama-4's iRoPE but does not disentangle the separate effects of interleaved NoPE layers and RoPE frequency scaling. An ablation study comparing would clarify which component drives the observed entropy shifts and performance outcomes.

**Questions:**

1. The sequence lengths in the paper are mostly set to 4096. Have you tried any other lengths, or does this length yield the most prominent effect?
2. An in-depth analysis of the FE features (such as Figure 2) is mainly conducted for "head 0". Is it necessary to systematically extract samples from different heads at different layers to observe whether the conclusions are consistent?
3. The downstream evaluation relies on benchmarks (e.g., HellaSwag, MMLU) that primarily evaluate general knowledge and reasoning. Could the "no significant difference" conclusion be undermined on other tasks specifically designed to assess long-context understanding?

---

> ### Author Response · Authors · 2025-11-22
> **Author Response (1/2)**
>
> Thank you very much for reviewing our paper. We sincerely appreciate your valuable feedback. Please find our responses below. All revisions made to the manuscript are highlighted in red.
>
> > **(Weakness 1) Experiments are conducted primarily on the Wikitext-103 dataset. To fully support the claim that FE is a "general diagnostic," the analysis should include diverse text datasets to verify whether the findings generalize beyond specific tasks.**
>
> Thank you for the suggestion. **We have added an analysis using the C4 and PG19 datasets in Appendix Section F2.** These results indicate that both SpectrumFE and SequenceFE consistently can capture their respective structural properties, and these properties do not shift when the input sequence is changed.
>
> > **(Weakness 2)The paper establishes correlations between FE values and model behaviors but does not rigorously prove causality. For instance, while attenuating low-SequenceFE dimensions doesn't hurt performance, it remains unclear if this is because they are truly redundant or if the model compensates via other mechanisms. A more controlled ablation would strengthen the causal claim.**
>
> Thank you for pointing this out. **We added this limitation in Appendix (Appendix Section D).**
>
> Our intervention selectively attenuates *sets* of dimensions identified by FE, so the model may redistribute representational load within the subspace.
> Our goal in this work is to provide a unifying *analysis* of RoPE frequency usage rather than to establish a complete causal theory. Accordingly, the weighted-RoPE experiments are intended, first-order interventions that probe whether FE-selected dimensions behave differently. They suggest functional differences between SpectrumFE and SequenceFE dimensions, but we do not claim they constitute a full causal identification.
>
> > **(Weakness 3) The paper studies Llama-4's iRoPE but does not disentangle the separate effects of interleaved NoPE layers and RoPE frequency scaling. An ablation study comparing would clarify which component drives the observed entropy shifts and performance outcomes.**
>
> **We conducted experiments training models from scratch with and without RoPE.** **Details are provided in Appendix G.** Training was performed on a 16-layer model. Results from Spectrum FE and Sequence FE for both RoPE-only and NoPE-only models revealed the following characteristics:
>
> - **NoPE**
>     - Thin frequency bands are visible only in some heads of shallow layers
>     - Frequency bands becomes non-retaining completely in deep layers
>     - No periodic structures are observed at all
> - **RoPE**
>     - Thin frequency bands are consistently present across all layers and all heads
>     - Strong periodic structures are not observed (they do not manifest as clear periodicity)
>
> Considering these results alongside Llama-4's behavior in Section 4's analysis, we found that NoPE alone cannot retain frequency bands, causing them to becomes non-retaining in deep layers. However, when combined with RoPE, NoPE actually enhances the frequency bands. However, we also observed that mixing NoPE and RoPE amplifies even the periodic components inherent to RoPE.
>
> These findings indicate that mixing RoPE and NoPE involves a tradeoff between the **advantage of band enhancement** and the **side effect of amplifying unwanted periodic components**.

---

> ### Author Response · Authors · 2025-11-22
> **Author Response (2/2)**
>
> > **(Question 1) The sequence lengths in the paper are mostly set to 4096. Have you tried any other lengths, or does this length yield the most prominent effect?**
>
> **We evaluated context lengths of 2048, 4096, and 8192, with results reported in Appendix F1.**
> The FE patterns remained similar across all lengths.
> We also conducted the same set of analyses on the C4 and PG19 datasets at these context lengths.
>
> > **(Question 2) An in-depth analysis of the FE features (such as Figure 2) is mainly conducted for "head 0". Is it necessary to systematically extract samples from different heads at different layers to observe whether the conclusions are consistent?**
>
> Thank you for your question. **The results for all heads are already shown in Figure 3, and the layer-wise evaluations are provided in Figures 6 and 7.** Although there are layer-specific differences, the head-wise behavior follows the same general patterns: the locations of the frequency bands and periodic dimensions vary across heads, but their overall characteristics remain consistent.
>
> > **(Question 3) The downstream evaluation relies on benchmarks (e.g., HellaSwag, MMLU) that primarily evaluate general knowledge and reasoning. Could the "no significant difference" conclusion be undermined on other tasks specifically designed to assess long-context understanding?**
>
> Thank you for your question. **We also evaluate long-context behavior using the Needle-in-a-Haystack (NIAH) task, with results given in Appendix E.**
> We use NIAH because it is a standard benchmark for position-interpolation–based context extension and is particularly sensitive to positional encoding and context length.
> While Llama-4 shows minor improvements under our intervention, the overall effect is small and not statistically significant.
>
> ---
>
> We sincerely thank you once again for your thoughtful and valuable comments on our paper. If you have any further questions or requests, we would be more than happy to address them.

---

> > ### Comment · Reviewer_dKd1 · 2025-11-26
> > **Reviewer response**
> >
> > Dear author:
> >     Thank you for the detailed explanation. Your reply has resolved my confusion. I will maintain this score. In my opinion, this is a very meaningful initiative, and I hope it will be accepted.
> > Best wishes

---

> > > ### Author Response · Authors · 2025-11-27
> > >
> > > Thank you very much for your kind follow-up and for taking the time to review our work. We sincerely appreciate your thoughtful evaluation.

---

### Official Review · Reviewer_USX3 · 2025-10-30

**Soundness:** 3
**Presentation:** 3
**Contribution:** 2
**Rating:** 4
**Confidence:** 3

**Summary:**

The paper introduces Frequency Entropy as a metric to study the utilization of RoPE frequency dimensions. The authors perform experiments on Llama-4 to compare the periodicity of RoPE and NoPE layers. The authors introduce Spectrum Frequency Entropy and Sequence Frequency Entropy to evaluate both which frequency components are present and how periodic the energy fluctuations are in each dimension.

**Strengths:**

- The methodology is interesting and provides a quantitative way to study RoPE utilization, as compared to prior more qualitative works
- The figures provide nice visualizations of the analysis and results
- The weighted RoPE experiments provide interesting insights on functional relevance, showing that suppressing low-Spectrum-FE dimensions worsens performance.

**Weaknesses:**

- The experiments are done on LLama-4 which interleaves RoPE and NoPE layers, so comparisons between RoPE and NoPE are not layer-matched. It would be interesting to see even at a smaller scale the differences between FE for a model trained with and without RoPE at a specific layer.
- The takeaways to me are not clear. In particular:
- The SpectrumFE results for RoPE and NoPE seem quite similar at later layers. It seems like the main difference is at the earliest layers. Do you have insights on the impact of this early-layer difference? Would a model with no RoPE layers exhibit the same late-layer behavior?
- What are the practical takeaways of the Weighted RoPE intervention?

**Questions:**

See weaknesses. In addition, prior work mentions that ablating low-frequency RoPE dimensions impacts long-context performance. I'd be interested to see the Weighted RoPE intervention in a similar setting to see if you could provide additional insights to long-context.

---

> ### Author Response · Authors · 2025-11-22
> **Author Response**
>
> Thank you very much for reviewing our paper. We sincerely appreciate your valuable feedback. Please find our responses below. All revisions made to the manuscript are highlighted in red.
>
> > **(Weakness 1) The experiments are done on LLama-4 which interleaves RoPE and NoPE layers, so comparisons between RoPE and NoPE are not layer-matched. It would be interesting to see even at a smaller scale the differences between FE for a model trained with and without RoPE at a specific layer.**
>
> Thank you for pointing that out.
> **We conducted experiments training models from scratch with and without RoPE (Appendix Section G).** Training was performed on a 16-layer model. Results from Spectrum FE and Sequence FE for both RoPE-only and NoPE-only models revealed the following characteristics:
>
> - **NoPE**
>     - Thin frequency bands are visible only in some heads of shallow layers
>     - Frequency bands becomes non-retaining completely in deep layers
>     - No periodic structures are observed at all
> - **RoPE**
>     - Thin frequency bands are consistently present across all layers and all heads
>     - Strong periodic structures are not observed (they do not manifest as clear periodicity)
>
> Considering these results alongside Llama-4's behavior in Section 4's analysis, we found that NoPE alone cannot retain frequency bands, causing them to becomes non-retaining in deep layers. However, when combined with RoPE, NoPE actually enhances the frequency bands. However, we also observed that mixing NoPE and RoPE amplifies even the high-frequency bands inherent to RoPE.
> These findings indicate that mixing RoPE and NoPE involves a tradeoff between the **advantage of band enhancement** and the **side effect of amplifying unwanted periodic components**.
>
> > **(Weakness 2) The SpectrumFE results for RoPE and NoPE seem quite similar at later layers. It seems like the main difference is at the earliest layers. Do you have insights on the impact of this early-layer difference? Would a model with no RoPE layers exhibit the same late-layer behavior?**
>
> As discussed in our response to Weakness 1, the targeted experiments (Section G) we conducted directly address this concern as well.
> For the RoPE-only model, the frequency-band structure appears consistently across both early and late layers, with only modest layer-wise shifts. This confirms that RoPE injects a stable frequency gradient from the beginning of training, and the later layers preserve this structure.
> In contrast, the NoPE-only model shows a different pattern: a weak band-like structure emerges in the earliest layers but this structure dissipates in the deeper layers.  This becomes non-retainingance of the band indicates that, without RoPE’s explicit rotational frequency encoding, the model does not maintain a stable frequency band as depth increases.
>
> > **(Weakness 3) What are the practical takeaways of the Weighted RoPE intervention?**
>
> This controlled intervention by weighted RoPE yields clear practical insights:
>
> - Suppressing low-SpectrumFE pairs (frequency-band components) consistently *harms* perplexity → these pairs carry important information for model performance.
> - Suppressing low-SequenceFE pairs (highly periodic components) leaves perplexity and downstream accuracy unchanged → these pairs are largely *redundant*.
>
> Thus practitioners can:
>
> (1) identify redundant RoPE dimensions for pruning or compression;
>
> (2) detect task-relevant frequency bands;
>
> (3) diagnose or design positional schemes by observing which bands are actually used.
>
> Weighted RoPE therefore serves as a diagnostic tool that transforms FE from a correlational descriptor into a practically actionable signal.
>
> > **(Question) See weaknesses. In addition, prior work mentions that ablating low-frequency RoPE dimensions impacts long-context performance. I'd be interested to see the Weighted RoPE intervention in a similar setting to see if you could provide additional insights to long-context.**
>
> Thank you for your question.
> **We also conduct long-context evaluations using the Needle-in-a-Haystack task, as reported in Appendix E.** Results similar to those in Section 5 were obtained. Although we observe slight improvements for some models, the overall differences remain small.
>
> ---
>
> We sincerely thank you once again for your thoughtful and valuable comments on our paper. If you have any further questions or requests, we would be more than happy to address them.

---

> > ### Comment · Reviewer_USX3 · 2025-11-26
> >
> > Thank you for the detailed rebuttal. I have a follow up question on the Appendix G experiments. I am quite confused on the training setting, specifically these details:
> > > The number of training epochs is 205, and the batch size is 9216. The learning rate was set to
> > 1.0, and the learning process was updated by 1e-7 every 16,000 steps. We used the WikiText-103
> > dataset (Merity et al., 2017), which consists of over 103 million tokens of English Wikipedia articles.
> >
> > This does not seem like a standard training setting, specifically this is many epochs over a very small dataset and learning rate 1.0 is much higher than standard practice. I am also confused by what is meant by "the learning process was updated by 1e-7 every 16,000 steps." Could you please clarify the details on this training setting? Some specific questions:
> > - what optimizer are you using?
> > - what is meant by the learning process being updated by 1e-7?
> > - is batch size 9216 in terms of tokens or sequences? what is the sequence length?

---

> ### Author Response · Authors · 2025-11-27
>
> Thank you for the follow-up question. Some parts of our earlier description were imprecise, and we appreciate the opportunity to clarify the exact configuration. The setup follows the small-scale diagnostic configuration used in prior work [1, 2], including the RoPE/NoPE comparison in [2]. To ensure reproducibility and comparability, we use the same Fairseq implementation and hyperparameters as in [1]. The corrected configuration is as follows.
>
> ### **Optimizer**
>
> We use Nesterov’s Accelerated Gradient (NAG) with momentum 0.99, following the Fairseq `nag` optimizer in [1, 2].
>
> ### **Learning rate schedule**
>
> The previous sentence “the learning process was updated by 1e-7 every 16,000 steps” was incorrect.
>
> The correct description is:
>
> - A cosine learning rate scheduler is used.
> - The learning rate is initialized at 1e-7
> - It is linearly warmed up to 1.0 over the first 16,000 updates
> - After warmup, the learning rate follows a cosine decay down to a minimum LR of 1e-4.
>
> ### **Batch size definition**
>
> The value 9216 corresponds to the maximum number of tokens per batch per GPU. Sequence length is  512 tokens.  Per-GPU batch is 9216 tokens.
>
> ### **Epoch calculation**
>
> We train for 286k updates, which corresponds to approximately 205 epochs on WikiText-103.
>
>
> We believe that this is a standard choice in prior work [1, 2], as the goal there is to observe structural differences in positional encoding behavior. These parameters exactly reproduce the small-scale Fairseq training recipe used in [1, 2]. We hope this clarifies the training setup. We have also revised the explanatory text in the paper (L1473-1481, blue text). We sincerely apologize for the earlier ambiguity, and we appreciate your careful reading and the opportunity to correct these descriptions. If you have any further questions, we would be more than happy to provide additional details.
>
> ---
>
> [1] Press+, Train Short, Test Long: Attention with Linear Biases Enables Input Length Extrapolation (ICLR2022) https://github.com/ofirpress/attention_with_linear_biases
>
> [2] Oka+, Wavelet-based Positional Representation for Long Context (ICLR2025)

---

### Official Review · Reviewer_G1Jc · 2025-10-31

**Soundness:** 3
**Presentation:** 4
**Contribution:** 3
**Rating:** 8
**Confidence:** 3

**Summary:**

This paper provides an analysis of RoPE in terms of Frequency Entropy (FE), or the Shannon entropy of the power spectrum. Two measures are used in this analysis: Spectrum FE which captures the active components and Sequence FE which quantifies the regularity of energy in each dimension. These quantities are normalized to provide scale-free measures. An analysis of RoPE in Llama-4 is presented. This analysis shows that RoPE and NoPE layers behave differently, but that earlier NoPE layers show bands under Spectrum FE similar to RoPE. This effect disappears at deeper layers. The paper also describes experiments to test the redundancy of RoPE dimensions with outlier FE by down-weighting them. Results show that low entropy dimensions (frequency bands) are important to model perplexity, though downstream tasks show similar performance between standard and weighted models.

**Strengths:**

The paper provides an interesting, theoretically grounded analysis of RoPE embeddings. It provides new insights into the contribution of RoPE on model performance, specifically distinguishing between periodic signals and high energy bands. The explanations are clear and the experimental evidence helps to solidify the analysis.

**Weaknesses:**

The choice to study only Llama-4 reduces the impact and generality of this paper. It is noted that NoPE layers in this model may attenuate periodic structures and emphasize frequency bands. If I understand, Spectrum and Sequence FE analysis has not been applied to a RoPE-only model like Gemma 7B or Llama-3. This would help to clarify the effect of NoPE in Llama 4.

**Questions:**

Did you consider attenuating outlying low or high-frequency components only? This could help strengthen the case that these specific component regions are not responsible for model performance.

---

> ### Author Response · Authors · 2025-11-22
> **Author Response**
>
> Thank you very much for reviewing our paper. We sincerely appreciate your valuable feedback. Please find our responses below. All revisions made to the manuscript are highlighted in red.
>
> > **(Weaknesses) The choice to study only Llama-4 reduces the impact and generality of this paper. It is noted that NoPE layers in this model may attenuate periodic structures and emphasize frequency bands. If I understand, Spectrum and Sequence FE analysis has not been applied to a RoPE-only model like Gemma 7B or Llama-3. This would help to clarify the effect of NoPE in Llama 4.**
>
> Thank you for raising this concern. We would like to clarify that our analysis is not limited to Llama-4. Appendix C provides full SpectrumFE and SequenceFE results for **Gemma-7B, Llama-3-8B, and Qwen-7B,** all of which employ *pure RoPE without NoPE layers*. The description is provided in the footnote on L268-269.
>
> Here is a brief summary:
>
>  - Frequency bands consistently appear across models, but their locations are model-dependent.
>  - SequenceFE remains high and clear periodic components are absent. Attenuating or pruning these periodic dimensions results in only a minor performance drop, consistent with downstream results in Section 5.3.
>  - Frequency bands appear in most heads, and reducing the contribution of band dimensions degrades performance (Section 5.2).
>
> Furthermore, we conducted small-scale pretraining on NoPE-only and RoPE-only models and compared them (Appendix Section G).
> Results from Spectrum FE and Sequence FE for both RoPE-only and NoPE-only models revealed the following characteristics:
>
> - **NoPE**
>     - Thin frequency bands are visible only in some heads of shallow layers
>     - Frequency bands becomes non-retaining in deep layers
>     - No periodic structures are observed at all
> - **RoPE**
>     - Thin frequency bands are consistently present across all layers and all heads
>     - Strong periodic structures are not observed (they do not manifest as clear periodicity)
>
> Considering these results alongside Llama-4's behavior in Section 4's analysis, we found that NoPE alone cannot retain frequency bands, causing them to becomes non-retaining in deep layers. However, when combined with RoPE, NoPE actually enhances the frequency bands. However, we also observed that mixing NoPE and RoPE amplifies even the periodic component inherent to RoPE.
>
> These findings indicate that mixing RoPE and NoPE involves a tradeoff between the **advantage of band enhancement** and the **side effect of amplifying unwanted periodic components**.
>
> > **(Questions) Did you consider attenuating outlying low or high-frequency components only? This could help strengthen the case that these specific component regions are not responsible for model performance.**
>
> Thank you for the suggestion. **We have already conducted this type of intervention through our weighted RoPE method, and Section 5** reports the perplexity results as well as evaluations on HellaSwag, TruthfulQA, and MMLU. **In addition, Appendix E provides long-context evaluations using the Needle-in-a-Haystack task.** Although we observe slight improvements for some models, the overall differences remain small.
>
> From the results, SpectrumFE-identified frequency band clearly matter for the model, while SequenceFE-identified periodic components contribute little.
> Overall, the experiments show that performance is not driven by absolute “low” or “high” frequencies, but by model-specific frequency bands whose locations vary across heads and layers.
>
> ---
>
> We sincerely thank you once again for your thoughtful and valuable comments on our paper. If you have any further questions or requests, we would be more than happy to address them.

---

### Meta-Review · Area_Chair_w7bT · 2025-12-25

**Summary:**

This paper proposes Frequency Entropy (FE) as a novel metric to analyze the frequency structure of Rotary Position Embeddings (RoPE), addressing prior conflicting findings on high- and low-frequency dimensions. Reviewers raised concerns regarding the generality of experiments initially focused on Llama-4, the layer-matched comparison between RoPE and NoPE, the practical utility of FE, the rigor of causal claims, and the diversity of evaluated datasets and contexts. After careful consideration of the paper’s contributions, the detailed rebuttal, and supplementary experiments, I intend to accept this submission.

**Reviewer Concerns:**

Most key concerns raised by reviewers have been effectively addressed through the authors’ rebuttal and supplementary analyses. The generality concern was resolved by extending experiments to RoPE-only models like Gemma-7B, Llama-3-8B, and Qwen7B, while layer-matched comparisons between RoPE and NoPE were supplemented by training small-scale models from scratch. Questions about dataset and context length generality were answered by adding analyses on C4, PG19 datasets, and multiple context lengths (2048, 4096, 8192). The practical value of FE was clarified by outlining applications in model compression, architectural tuning, and training diagnostics, and long-context performance was verified via the Needle-in-a-Haystack task. Outstanding concerns are limited to the inherent difficulty of fully establishing causal relationships (acknowledged by the authors as a limitation) and residual doubts about FE’s actionable impact, which do not undermine the core contribution of the framework.

**Reviewer Scores:**

Overall, the reviewers’ scores would likely shift positively following the authors’ comprehensive rebuttal and supplementary experiments. Initial lower scores stemmed from unaddressed concerns about generality, experimental design, and practical utility—all of which have been substantially mitigated. Reviewers who previously expressed reservations about the work’s scope and rigor would find their key questions answered, while those already supporting acceptance would maintain their positive assessments. This is a meaningful and valuable work that fills a gap in understanding RoPE’s internal frequency dynamics, and I encourage the authors to revise the manuscript in line with the reviewers’ comments and their own rebuttal to further strengthen the presentation and clarity of the findings.

---

### Decision · Program_Chairs · 2026-01-26

Accept (Poster)